# Glutaminolysis provides nucleotides and amino acids to regulate osteoclast differentiation in mice

Guoli Hu[1], Yilin Yu[1], Yinshi Ren[2,3], Robert J Tower [ID] [4,5], Guo-Fang Zhang [ID] [6,7] & Courtney M Karner [ID] [1,5 ✉]

## Abstract

**Osteoclasts are bone resorbing cells that are essential to maintain skeletal integrity and function. While many of the growth factors and molecular signals that govern osteoclastogenesis are well studied, how the metabolome changes during osteoclastogenesis is unknown. Using a multifaceted approach, we identified a metabolomic signature of osteoclast differentiation consisting of increased amino acid and nucleotide metabolism. Maintenance of the osteoclast metabolic signature is governed by elevated glutaminolysis. Mechanistically, glutaminolysis provides amino acids and nucleotides which are essential for osteoclast differentiation and bone resorption in vitro. Genetic experiments in mice found that glutaminolysis is essential for osteoclastogenesis and bone resorption in vivo. Highlighting the therapeutic implications of these findings, inhibiting glutaminolysis using CB-839 prevented ovariectomy induced bone loss in mice. Collectively, our data provide strong genetic and pharmacological evidence that glutaminolysis is essential to regulate osteoclast metabolism, promote osteoclastogenesis and modulate bone resorption in mice.**

**Keywords** Glutaminolysis; Amino Acids; Nucleotides; Osteoclast; Osteoporosis
**Subject Categories** Metabolism; Musculoskeletal System

## Introduction

Osteoclasts (OC) are giant, multinucleated cells that specialize in the resorption of calcified bone matrix through acidic and enzymatic processes (Boyle et al, 2003). Bone resorption is essential to maintain skeletal integrity and function. OC differentiate from myeloid progenitor cells through a multistep process that is initiated by macrophage colony-stimulating factor (M-CSF) and receptor activator of nuclear factor-κB ligand (RANKL) (Boyle et al, 2003; Kikuta and Ishii, 2013; Teitelbaum and Ross, 2003; Zhao and Ivashkiv, 2011). Binding of M-CSF and RANKL to their

receptors, c-Fms and RANK, respectively (Lee and Lorenzo, 2006; Wada et al, 2006), activates NFATc1 to drive the expression of osteoclast specific genes (Amarasekara et al, 2018; Hamerman et al, 2006; Humphrey et al, 2006; Koga et al, 2004), including *Acp5* (TRAP), *Dc-stamp*, *Atp6v0d2*, *Itgb3* and *Ctsk* (Grigoriadis et al, 1994; Kim et al, 2002; Takayanagi et al, 2002). OC then undergo cell fusion, polarization and activation characterized by cytoskeletal reorganization, formation of F-actin sealing zone and the ruffled membrane, and polarized secretion of acids and proteolytic enzymes into the resorbing lacunae (Kikuta and Ishii, 2013; Wang et al, 2003). Excessive OC generation and/or bone resorption underlie many bone and joint diseases including postmenopausal osteoporosis, periprosthetic osteolysis, Paget's bone disease, rheumatoid arthritis (RA) and osteolytic bone metastasis (Choi et al, 2009; Macedo et al, 2017; Sato and Takayanagi, 2006; Shaker, 2009; Teitelbaum, 2006).

Osteoclastogenesis and bone resorption are energetically demanding and likely burden osteoclasts with constantly evolving metabolic needs. Recent studies have begun to interrogate osteoclast metabolism and have described several changes in nutrient uptake and metabolic activity that occur during osteoclastogenesis. These changes include increased glucose uptake and glycolysis, increased fatty acid oxidation, increased mitochondrial biogenesis and activity, and increased glutamine uptake (Arnett and Orriss, 2018; Da et al, 2021; Indo et al, 2013; Ishii et al, 2009; Kim et al, 2021; Kushwaha et al, 2022; Lemma et al, 2016; Li et al, 2020; Peng et al, 2024; Song et al, 2023; Zeng et al, 2015; Zhang et al, 2018). Moreover, these studies demonstrated that glucose and fatty acids are preferred energetic substrates in osteoclasts (Kushwaha et al, 2022; Li et al, 2020; Song et al, 2023). However, genetic ablation of either glucose uptake or fatty acid oxidation in osteoclast precursors using *LysmCre* results in moderate sexually dimorphic bone phenotypes in vivo, suggesting other metabolic pathways play a more dominant role (Kushwaha et al, 2022; Li et al, 2020; Song et al, 2023). Recent studies found serine synthesis, which provides α-ketoglutarate (α-KG) to regulate *Nfatc1* expression, is another key metabolic pathway for osteoclastogenesis (Stegen et al, 2024). Along those same lines, the deubiquitinating enzyme BAP1 was found to be critical for osteoclast function as loss of BAP1 in osteoclasts resulted in metabolic reprogramming, namely reduced mitochondrial respiration, increased glutathione (GSH) and altered abundance of several amino acids (Rohatgi et al, 2023). Despite

[1]Department of Internal Medicine, University of Texas Southwestern Medical Center, Dallas, TX 75390, USA. [2]Center for Excellence in Hip Disorders, Texas Scottish Rite Hospital for Children, Dallas, TX 75219, USA. [3]Department of Orthopedic Surgery, University of Texas Southwestern Medical Center, Dallas, TX 75390, USA. [4]Department of Surgery, University of Texas Southwestern Medical Center, Dallas, TX 75390, USA. [5]Charles and Jane Pak Center for Mineral Metabolism and Clinical Research, University of Texas Southwestern Medical Center, Dallas, TX 75390, USA. [6]Department of Medicine, Division of Endocrinology, Metabolism Nutrition, Duke University Medical Center, Durham, NC 27701, USA. [7]Sarah W. Stedman Nutrition and Metabolism Center & Duke Molecular Physiology Institute, Duke University School of Medicine, Durham, NC 27701, USA. ✉E-mail: Courtney.Karner@UTSouthwestern.edu

these recent advances, it is unclear if or how the intracellular metabolome changes during osteoclastogenesis and how the metabolism of individual nutrients impacts the osteoclast metabolome and thus influences osteoclastogenesis and bone resorption.

Glutamine metabolism plays essential regulatory functions in many diverse cell types. For example, many cancer cells are glutamine addicted and use glutamine to support proliferation and biomass generation (DeBerardinis et al, 2007). In the context of bone, both osteoblasts and chondrocytes rely on glutamine metabolism for cell differentiation and matrix production. In osteoblast lineage cells, glutamine provides α-KG, amino acids and GSH to regulate proliferation, osteoblast differentiation, bone matrix production and redox homeostasis (Hu et al, 2023; Karner et al, 2015; Sharma et al, 2021; Stegen et al, 2021; Stegen et al, 2016; Yu et al, 2019). Chondrocytes use glutamine to provide acetyl-CoA, aspartate and GSH to regulate gene expression, matrix biosynthesis and redox homeostasis (Stegen et al, 2020; Zhang et al, 2022). In osteoclasts, exogenous glutamine is required for osteoclast differentiation in vitro, while RANKL stimulation increases the expression of the neutral amino acid transporter *Slc1a5* as well as *Gls*, the enzyme that catalyzes the first step in glutaminolysis (Indo et al, 2013; Peng et al, 2024). This suggests glutaminolysis may be important during osteoclastogenesis, however, the necessity of glutaminolysis and how this contributes to either osteoclast metabolism or osteoclast differentiation, fusion and/or function have not been investigated.

Here, we describe the metabolic signature of differentiating osteoclasts. This signature is enriched for metabolites associated with amino acid and nucleotide metabolism. We show that glutamine uptake and glutaminolysis are essential for osteoclastogenesis-associated metabolic reprogramming. Moreover, glutaminolysis is necessary and sufficient for osteoclastogenesis and bone resorption in vivo. Mechanistically, our data indicate that transamination of glutamine-derived glutamate provides amino acids and nucleotides to fuel osteoclast differentiation and bone resorption. Finally, inhibiting glutaminolysis using CB-839 protects mice from bone loss caused by estrogen deficiency, highlighting the therapeutic potential of targeting this pathway.

## Results

### Osteoclast metabolome is characterized by elevated amino acids and nucleotides

To gain insights about the metabolic mechanisms underlying osteoclastogenesis, we performed either RNA sequencing (RNA-seq) or LC-MS/MS to profile the transcriptomic or metabolomic changes of primary bone marrow macrophage (BMM) induced to undergo osteoclast differentiation in response to M-CSF and RANKL for up to 4 days (Fig. 1A,B). We identified 5284 differentially expressed genes (DEG) in pre-osteoclast (pOC) and 6545 DEG in mature osteoclast (mOC) compared to BMM. DEG enriched in mOC were associated with osteoclast differentiation, actin cytoskeletal organization and cell-cell fusion, whereas DEG enriched in BMM were associated with macrophage activation and differentiation (Fig. EV1A,B). Interestingly, genes associated with several amino acid and nucleotide metabolic processes were significantly enriched in mOC, including amino acid metabolism,

amino acid biosynthesis, nucleotide metabolism, nucleobase-containing compound biosynthesis and purine-containing compound biosynthesis among others (representative genes shown in Fig. 1C) (Fig. EV1B). Conversely, DEG enriched in BMM were associated with glucose and fatty acid metabolism, including glucose homeostasis and import, lipid transport, fatty acid metabolism and energy reserves (Fig. EV1A). LC-MS/MS identified 123 metabolites that differed significantly between BMM and pOC and 145 metabolites that differed significantly between BMM and mOC, respectively (Fig. EV1C). Analysis of the altered metabolites using MBROLE (Lopez-Ibanez et al, 2016) revealed a comprehensive metabolic signature of osteoclastogenesis. Consistent with the RNA-seq data, abundant metabolites in mOC were enriched for amino acid and nucleotide metabolic pathways (Fig. EV1D). Several nucleotide metabolites (exemplified by the purine nucleoside inosine and pyrimidine nucleoside uridine) were detected in BMM and increased substantially during osteoclastogenesis (Figs. 1D and EV1E,F). Many nucleotide metabolites, exemplified by the deoxyribonucleoside deoxyguanosine and the ribonucleoside cytidine, were not detected in BMM but were abundant in pOC and mOC (Figs. 1D and EV1G,H). These experiments were performed in media lacking ribonucleosides and deoxyribonucleosides, suggesting the increased intracellular abundance of these nucleotide metabolites is driven by de novo nucleotide biosynthesis. Similarly, the abundance of every amino acid except histidine and serine increased significantly during osteoclastogenesis (Fig. 1D). By comparison, metabolites that were abundant in BMM were associated with energetic and mitochondrial metabolism. These pathways included the mitochondrial electron transport chain, β-oxidation of various fatty acids and glycolysis among others (Fig. EV1I). Consistent with this, intracellular ATP levels were most abundant in BMM (Figs. 1D and EV1J).

The abundance of individual amino acids did not increase uniformly. For example, alanine, glutamine, and proline were each increased in abundance in both pOC and mOC (Figs. 1D and EV1K–M). However, the fold increase in proline abundance far surpassed either alanine or glutamine in mOC. By comparison, glutamate abundance was not changed in pOC but was increased in mOC (Fig. EV1N). Radiolabeled amino acid uptake assays found that BMM consumed glutamine at a greater rate compared to other amino acids (Fig. 1E). Moreover, while proline, glutamate and alanine consumption increased significantly, glutamine consumption far exceeded the other amino acids at all stages of osteoclastogenesis (Fig. 1F–I). Interestingly, glutamine uptake outpaced the increase in intracellular glutamine, whereas glutamate abundance was unchanged in pOC despite increased glutamate consumption (Figs. 1F,G and EV1L,N). Moreover, intracellular alanine abundance increased in pOC despite no change in alanine consumption (Figs. 1I and EV1K). These data suggest amino acid abundance is regulated by multiple factors including amino acid consumption, de novo amino acid synthesis and possibly amino acid catabolism. Further, glutamine demand far outpaces the demand for other amino acids during osteoclastogenesis.

### Glutaminolysis is essential to maintain amino acid and nucleotide abundance in osteoclasts

Osteoclasts substantially increase glutamine consumption during differentiation, despite glutamine being one of the most abundant intracellular amino acids in osteoclasts. This indicates osteoclasts have

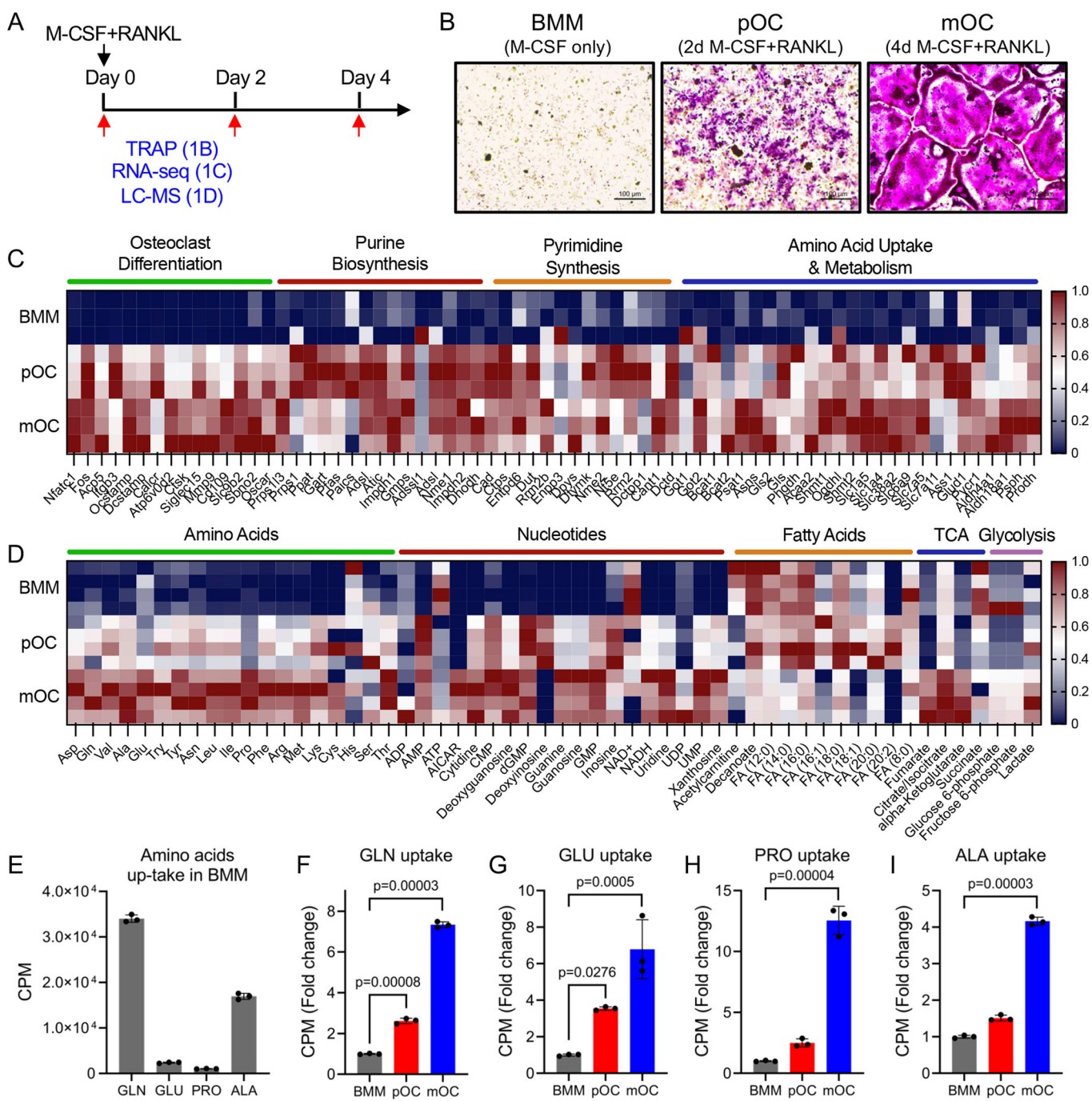

**Figure 1. Amino acid and nucleotide abundance increases during osteoclast differentiation.**

(**A**) Schematic depicting the experimental design. Primary bone marrow macrophages (BMMs) were stimulated with M-CSF and RANKL followed by TRAP staining, RNA-seq, or metabolomics after 0, 2, or 4 days of treatment. (**B**) TRAP staining of BMM stimulated with RANKL (40 ng/mL) for 0, 2, or 4 days ($n = 3$ cultures). Scale bar: 100 µm. (**C**) Heatmap showing differentially expressed genes in BMM, pOC, and mOC as determined by RNA-seq ($n = 3$). The expression for each gene (counts per million) was normalized on a Min Max Scale. (**D**) Heatmap showing relative abundance of metabolites in BMM, pOC, and mOC as determined by mass spectrometry ($n = 4$). The abundance of each metabolite was normalized on a Min Max Scale. (**E**) Graphical depiction of the basal consumption of 4 different radiolabeled amino acids in BMM ($n = 3$ independent experiments). (**F–I**) Graphical depiction of radiolabeled amino acid uptake assays in BMM, pOC, and mOC ($n = 3$ independent experiments). Data are shown as mean ± SD. One-way ANOVA (**F–I**). Source data are available online for this figure.

an acute demand for glutamine during differentiation. In addition to supporting protein synthesis, glutamine provides carbon and nitrogen for amino acid (e.g., aspartate, alanine, asparagine, arginine, proline, and branched-chain amino acids) and nucleotide (purines and pyrimidines) biosynthesis. This led us to hypothesize that glutaminolysis supports amino acid and nucleotide biosynthesis during osteoclast differentiation. Consistent with this, analysis of an existing scRNAseq dataset and western blot analyses found the expression of genes and proteins involved in glutamine uptake and metabolism into various amino acids and nucleotides are upregulated during osteoclastogenesis (Figs. 2A and EV2A–F) (Tsukasaki et al, 2020, Data Ref: Tsukasaki M, 2020). Functional assays found that GLS enzymatic activity increased significantly during osteoclastogenesis (Fig. 2B). Stable isotope tracing using [U-$^{13}$C]-glutamine (Fig. 2C) found the flux of glutamine changes during osteoclastogenesis. In BMM, glutamine carbon was enriched in glutamate, the tricarboxylic acid (TCA) cycle intermediates (e.g., α-KG, succinate, fumarate, malate, and citrate), aspartate and GSH (Figs. 2D–I and EV2G) with negligible enrichment in other amino acids (e.g., proline and alanine) (Fig. EV2G). Glutamine contribution to TCA cycle intermediates, aspartate and GSH was increased in pOC before declining slightly in mOC (Figs. 2F–I and EV2G). By comparison, glutamine contribution to glutamate and α-KG continued to increase throughout osteoclastogenesis (Fig. 2D,E). Thus, glutamine carbon fluxes primarily into α-KG, which is used for aspartate biosynthesis. GLS inhibition in mOC using Bis-2-(5-phenylacetamido-1,3,4-thiadiazol-2-yl) ethyl sulfide (BPTES) significantly reduced the intracellular abundance of α-KG, all amino acids, GSH and many nucleotides (Figs. 2J,K and EV2H). Analyses of the altered metabolites found amino acid pathways, purine and pyrimidine metabolism, TCA cycle, and GSH metabolism were significantly reduced upon GLS inhibition (Fig. EV2I). By comparison, no metabolic pathways were enriched upon BPTES treatment. Thus, glutaminolysis is essential to maintain homeostatic concentrations of amino acids and nucleotides in mature osteoclasts.

## Glutaminolysis is essential for osteoclast differentiation and bone resorption

We next sought to determine the role of GLS and glutaminolysis during osteoclastogenesis. Acute inhibition of GLS in BMM using BPTES disrupted osteoclastogenesis as exemplified by reductions in the number of TRAP$^+$ multinucleated osteoclasts and the number of nuclei per osteoclast (Fig. 3A,B). Notably, *Nfatc1* and *c-Fos* induction by RANKL was not affected (Fig. 3C,D), suggesting RANKL-RANK downstream signaling is intact. Rather, the reduction in the number of multinucleated osteoclasts was attributed to a defect in osteoclast fusion shown by decreased nuclei per cell, reduced expression of fusion genes (e.g., *Dc-stamp* and *Atp6v0d2*) and mature osteoclast marker genes (e.g., *Acp5*, *Itgb3*, and *Ctsk*), a failure to form the actin belt and defective bone resorption in the pit assay (Fig. 3B,D–F). Finally, acute GLS inhibition after fusion in mOC prevented bone resorption (Fig. 3G,H). These data indicate glutaminolysis is essential for osteoclast differentiation and bone resorption in vitro.

We next deleted a conditionally null (floxed) allele of *Gls* (*Gls$^{fl}$*) in myeloid lineage osteoclast precursors using *LysMCre*. GLS protein was efficiently depleted in both undifferentiated and differentiated *LysMCre;Gls$^{fl/fl}$* BMM and did not affect RANKL-induced NFATc1 accumulation (Fig. 4A). Despite this, *LysMCre;Gls$^{fl/fl}$* BMM had defects in cell fusion and were unable to differentiate into mOC as

evident from the presence of significantly fewer, smaller TRAP$^+$ osteoclasts that contained significantly fewer nuclei, reduced actin belt formation and reduced expression of osteoclast genes (Fig. 4B–F). At 4-months of age, both male and female *LysMCre;Gls$^{fl/fl}$* mice had significantly increased bone volume compared to wild-type littermate controls (Figs. 4G,H and EV3). Analysis of the μCT data found *LysMCre;Gls$^{fl/fl}$* mice had increased trabecular bone volume per tissue volume (BV/TV), trabecular number (Tb.N) and trabecular thickness (Tb.Th) and decreased trabecular separation (Tb.Sp) in both males and females at 4-months of age (Fig. EV3A,C). Cortical thickness (Ct.Th) was not affected in either males or females (Fig. EV3A,C). *LysMCre;Gls$^{fl/fl}$* mice had significantly fewer TRAP$^+$ osteoclasts per bone surface (Oc.N/BS) and decreased bone resorption as demonstrated by decreased serum TRAP5B and C-terminal telopeptide of collagen (CTX-1) (Fig. 4I–L). *LysMCre;Gls$^{fl/fl}$* mice had no difference in the number of osteoblasts per bone surface (Ob.N/BS) (Fig. 4M,N). Likewise, serum N-terminal pro-peptide of type 1 collagen (P1NP), a bone formation marker, was not affected in the *LysMCre;Gls$^{fl/fl}$* mice (Fig. 4O). We observed similar results in mice in which *Gls* was conditionally knocked out in osteoclast precursors using *Csfr1Cre$^{ERT2}$* at 1 month of age by giving tamoxifen (Appendix Fig. S1). Thus, *Gls* and glutaminolysis are required for osteoclast fusion, differentiation and bone resorption in vivo.

## Increasing glutaminolysis enhances osteoclastogenesis and bone resorption

Next, we sought to understand if increasing glutaminolysis is sufficient to enhance osteoclastogenesis. We generated *LysMCre;Rosa26$^{rtTA}$;tetO$^{Gls-t2a-mCherry}$* mice (denoted from hereon as *LysM;Gls$^{DOXON}$*) capable of expressing 2a-tagged GLS and mCherry in myeloid lineage cells in the presence of doxycycline (Appendix Fig. S2A). Western blot analyses found doxycycline increased total GLS and 2A-tagged GLS expression in *LysM;Gls$^{DOXON}$* BMM but not wild-type BMM (Fig. 5A). Increasing GLS expression in *LysM;Gls$^{DOXON}$* BMM accelerated osteoclast differentiation and enhanced bone resorption in the pit assay (Fig. 5B–F). This was evident by earlier and more robust induction of osteoclast fusion (e.g., *Atp6v0d2* and *dc-stamp*) and mature marker genes (e.g., *Acp5*, *Itgb3*, and *Ctsk*), increased mOC number, increased mOC area, and increased nuclei per mOC (Fig. 5B–F; Appendix Fig. S2B,C). Moreover, we observed the appearance of multinucleated TRAP$^+$ osteoclasts in *LysM;Gls$^{DOXON}$* BMM after 2 days of RANKL stimulation which were not evident until 3 days in wild-type BMM (Fig. 5B,C). μCT analyses found that increasing *Gls* expression in myeloid cells resulted in reduced bone mass (Fig. 5G,H). *LysM;Gls$^{DOXON}$* mice had significantly reduced BV/TV, Tb.N, and Tb.Th while Tb.Sp was increased compared to wild-type littermates following one month on doxycycline chow (Appendix Fig. S2D). Decreased bone mass in *LysM;Gls$^{DOXON}$* mice was attributed to increased bone resorption and increased osteoclast numbers as shown by serum TRAP5B, serum CTX-1 and TRAP staining, respectively (Fig. 5I–L). Histological analysis found *LysM;Gls$^{DOXON}$* mice had fewer trabeculae despite no change in osteoblast numbers or activity as shown by Ob.N/BS and serum P1NP levels (Fig. 5M–O). Together, these data indicate that increasing glutaminolysis in myeloid lineage cells accelerates osteoclast differentiation and enhances bone resorption in mice.

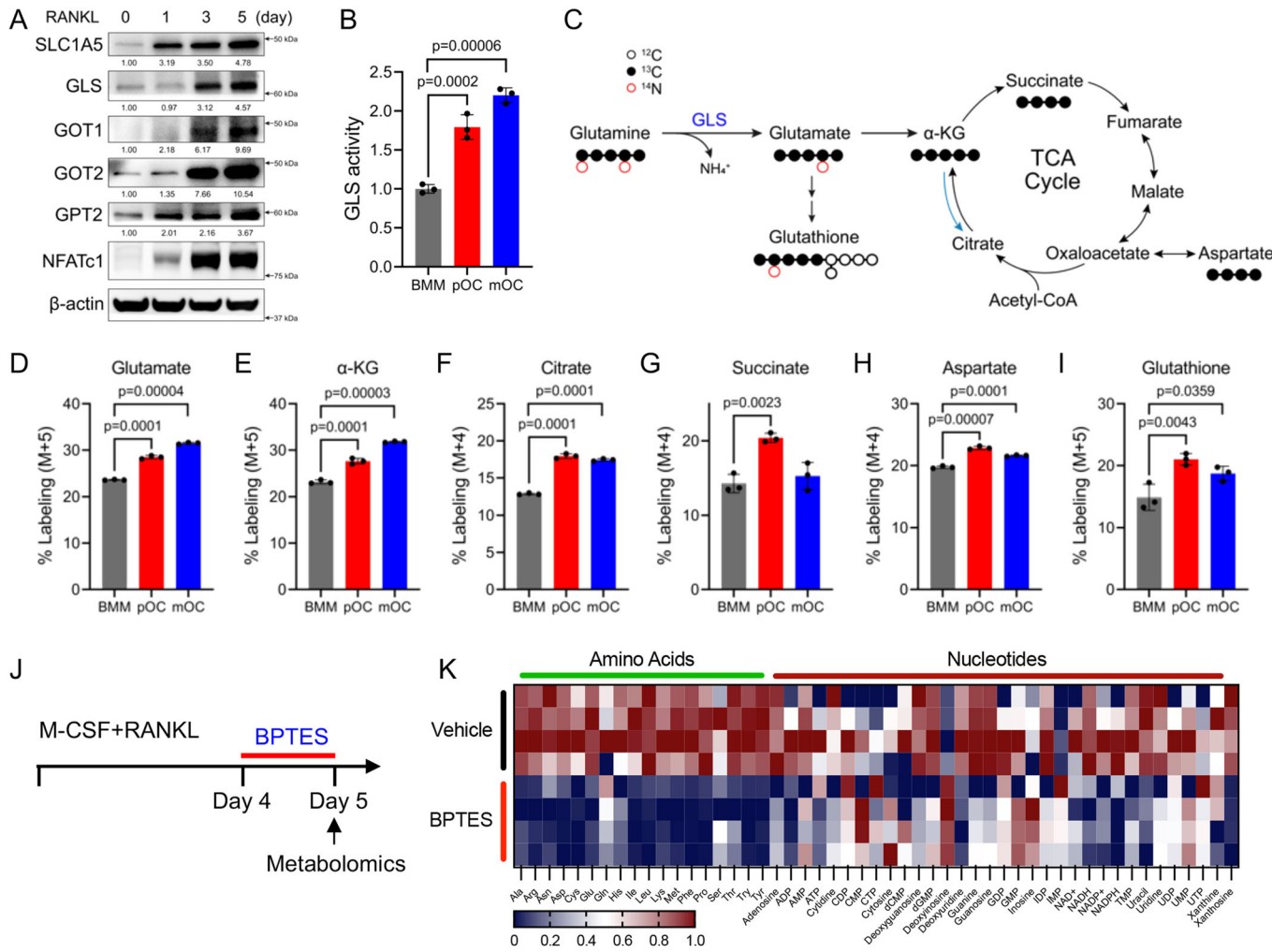

**Figure 2. RANKL stimulates glutaminolysis to support osteoclastogenesis-associated metabolic reprogramming.**

(A) Representative western blot images of glutamine metabolic enzymes in BMM stimulated with RANKL for the indicated time ($n = 3$). Protein expression normalized to β-actin. (B) Glutaminase activity in BMM, pOC, and mOC ($n = 3$ independent experiments). (C) Schematic depicting glutamine carbon metabolism and isotope tracing strategy. (D–I) Graphical depiction of percent labeling of glutamate, α-KG, citrate, succinate, aspartate, and glutathione from [U-$^{13}$C]-glutamine in BMM, pOC, and mOC ($n = 3$ independent experiments). (J) Schematic depicting the experimental design. GLS inhibitor BPTES (10 μM) was added at day 4 and metabolomics were performed 24 h later. (K) Heatmap showing relative abundance of metabolites in BPTES- or vehicle-treated mOC as determined by mass spectrometry ($n = 4$). Data are shown as mean ± SD. One-way ANOVA (B, D–I). Source data are available online for this figure.

## Glutamate transamination provides nucleotides and amino acids to regulate bone resorption

Glutaminolysis maintains the intracellular abundance of GSH, amino acids and nucleotides in mOC (Figs. 2 and EV2). Next, we sought to understand the role of glutamine-derived metabolites during osteoclast differentiation and bone resorption using rescue experiments. First, we focused on αKG, as 32% of αKG in mOC is derived from glutamine and others recently demonstrated the essential role for αKG in regulating NFATc1 expression (Stegen et al, 2024). Supplementation of glutamine-free media with cell-permeable dimethyl-αKG (DMαKG) resulted in the formation of a few TRAP$^+$ multinucleated mOC (Fig. EV4A–C). However, DMαKG was unable to rescue the expression of mOC genes (Fig. EV4D–H). We next supplemented BPTES-treated BMM with either nucleotides or non-essential amino acids (NEAA). Whereas

NEAA alone did not rescue the formation of TRAP$^+$ mOC, we did observe a few multinucleated mOC following the administration of nucleotides (Fig. 6A). By comparison, the combined administration of NEAA and nucleotides almost completely restored osteoclast formation following BPTES treatment (Fig. 6A). Consistent with a rescue of osteoclast differentiation, the combination of NEAA and nucleotides was able to restore actin belt formation, terminal osteoclast gene expression and bone resorption when GLS was inhibited (Figs. 6B–G and EV4I–N). Moreover, the combined administration of NEAA and nucleotides also rescued bone resorption in BPTES-treated mOC (Fig. EV4O–P). Together, glutaminolysis provides NEAA and nucleotides to govern osteoclast differentiation and bone resorption.

Next, we sought to understand how glutaminolysis supports amino acid and nucleotide synthesis in osteoclasts. The amino acid transaminases utilize the α nitrogen from glutamate for amino acid

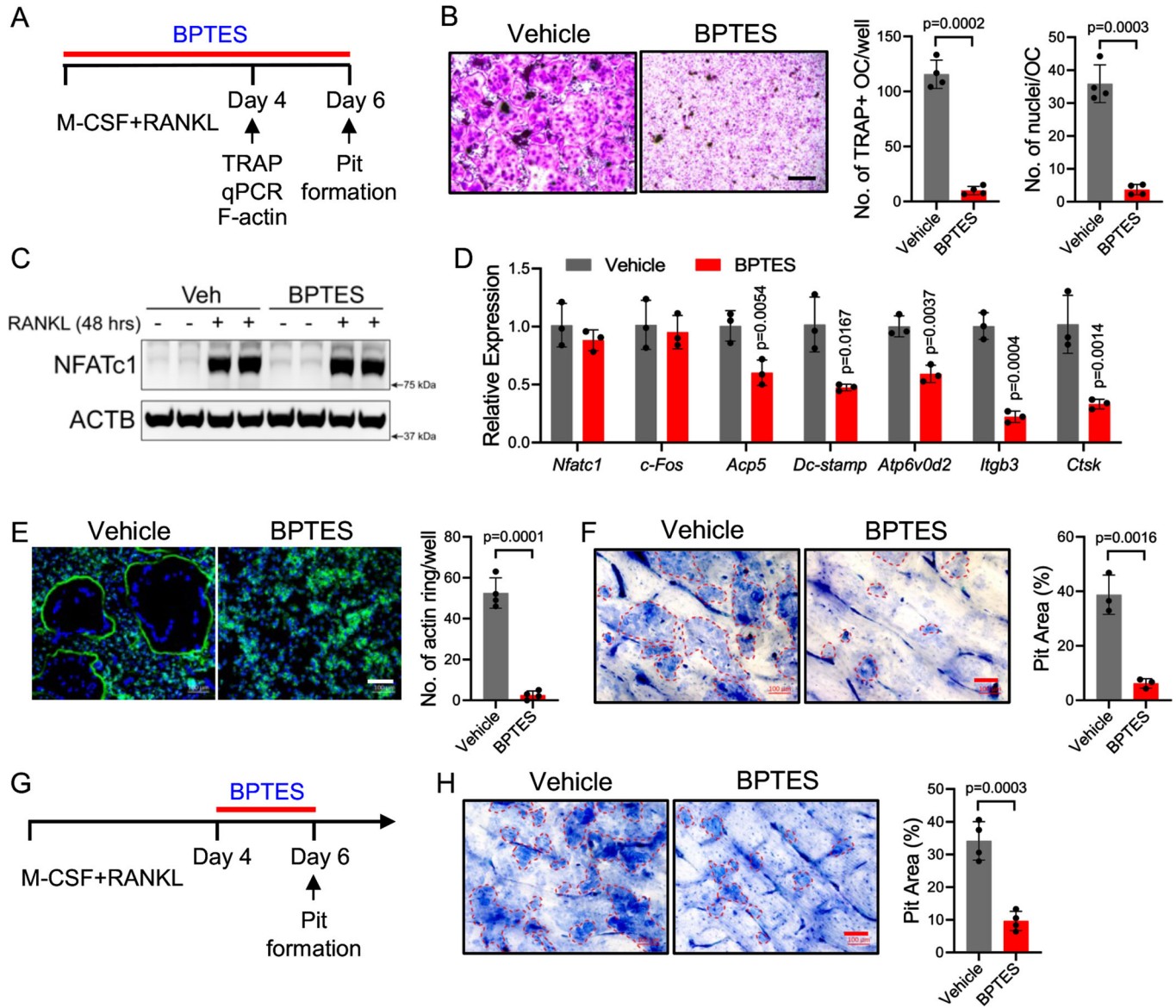

**Figure 3. Glutaminolysis is essential for osteoclast differentiation and bone resorption in vitro.**

(A) Schematic depicting the experimental design. BMM were induced with M-CSF and RANKL in the presence of the GLS inhibitor BPTES (10 μM). (B) TRAP staining showing the effect of BPTES on the formation of multinucleated TRAP-positive osteoclasts ($n = 4$ independent cultures). Scale bar: 500 μm. Quantification of the number of TRAP-positive multinucleated osteoclasts and the average number of nuclei per osteoclast is shown to the right. (C) Western blot analyses of the effect of BPTES on NTAFc1 induction by RANKL ($n = 4$). (D) qPCR analyses of the effect of BPTES on osteoclast gene expression ($n = 3$ independent experiments). (E) Phalloidin staining showing the effect of BPTES on actin belt formation ($n = 4$ independent cultures). Scale bar: 100 μm. (F) Resorption pit assay showing the effects of BPTES on bone resorption of bovine bone slices in vitro ($n = 3$ independent cultures). Scale bar: 100 μm. (G) Schematic depicting the experimental design for acute GLS inhibition in (H). BMM were cultured with M-CSF and RANKL on bovine bone slices. BPTES was added on day 4 and resorption pits were identified using toluidine blue at day 6. (H) Resorption pit assay showing that BPTES treatment beginning on day 4 reduces bone resorption in vitro ($n = 4$ independent cultures). Scale bar: 100 μm. Data are shown as mean ± SD. Two-tailed Student's unpaired $t$ test (B, D, E, F, H). Source data are available online for this figure.

biosynthesis (e.g., aspartate, alanine, serine, or branched-chain amino acids (BCAAs) valine, leucine, and isoleucine) (Fig. 7A). Stable isotope tracing using a glutamine tracer labeled with $^{15}$N on the α nitrogen ([α-$^{15}$N]-glutamine) found glutamine α nitrogen was enriched in glutamate and aspartate suggesting high glutamate oxaloacetate transaminase (GOT) activity in BMM (Fig. 7B,C). Glutamine α nitrogen was found in lower amounts in alanine, leucine, isoleucine, and valine, whereas serine or glycine had little labeling (Fig. 7D–I).

Glutamine α nitrogen enrichment increased throughout osteoclastogenesis in all amino acids except for valine which was static, and serine and glycine which declined to undetectable levels (Fig. 7G–I). We next treated mOC with the pan amino acid transaminase inhibitor aminooxyacetate (AOA) and evaluated the effects on the metabolome (Appendix Fig. S3A). AOA treatment significantly reduced 114 intracellular metabolites including many amino acids and nucleotides (Fig. 7J; Appendix Fig. S3B). Analysis of the altered metabolites found

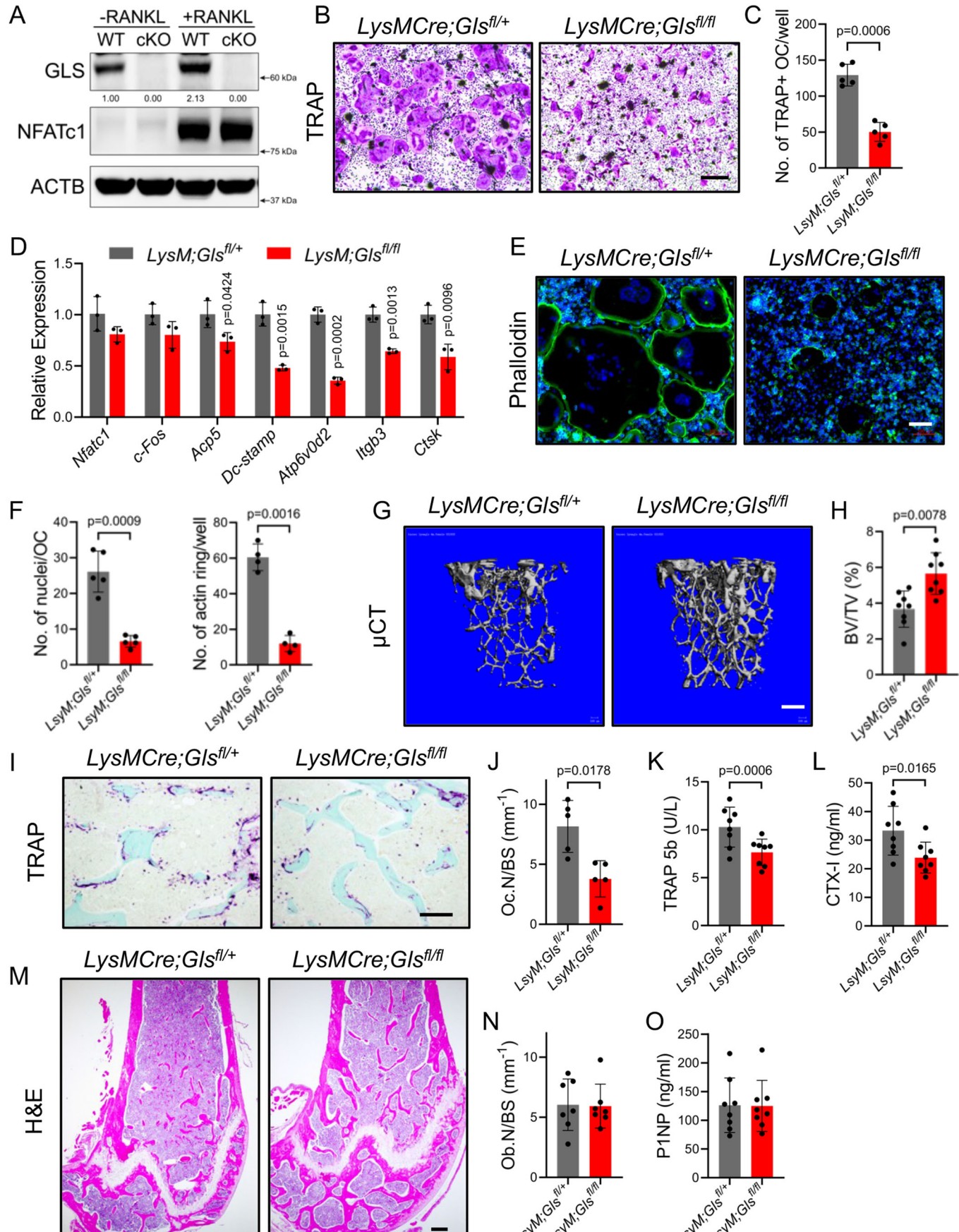

Figure 4.  **LysMCre;Gls$^{fl/fl}$ mice have increased bone mass due to reduced osteoclast numbers and bone resorption.**

(A) Western blot analysis of LysMCre;Gls$^{fl/+}$ (WT) and LysMCre;Gls$^{fl/fl}$ (cKO) BMM cultured with or without RANKL for 72 h ($n = 3$). (B–D) TRAP staining (B, C) or qPCR (D) of primary BMM isolated from LysMCre;Gls$^{fl/+}$ and LysM;Gls$^{fl/fl}$ female mice and cultured with RANKL for 4 days ($n = 5$ independent experiments). Scale bar: 500 μm. (C) Quantification of the number of TRAP-positive multinucleated osteoclasts per well ($n = 5$). (D) qPCR showing reduced induction of osteoclast genes in LysM;Gls$^{fl/fl}$ BMM induced with RANKL for 4 days ($n = 3$ independent experiments). (E) Phalloidin staining showing reduced actin belt formation in LysM;Gls$^{fl/fl}$ BMM ($n = 4$). Scale bar: 100 μm. (F) Quantification of the number of nuclei per osteoclast ($n = 5$) and the number of actin belts ($n = 4$). (G–H) Representative μCT (G) and calculated bone volume per tissue volume (BV/TV) (H) from the distal femur of 4-month-old LysMCre;Gls$^{fl/+}$ and LysM;Gls$^{fl/fl}$ female mice ($n = 8$ mice). Scale bar: 200 μm. (I–J) Representative TRAP staining (I) and quantification of osteoclast number per bone surface (Oc.N/BS) (J) from distal femurs from 4-month-old LysMCre;Gls$^{fl/+}$ and LysM;Gls$^{fl/fl}$ female mice ($n = 5$). Scale bar: 100 μm. (K–L) Serum ELISA for TRAP 5b (K) and CTX-I (L) from 4-month-old LysMCre;Gls$^{fl/+}$ and LysM;Gls$^{fl/fl}$ female mice ($n = 8$). (M, N) Representative H&E staining ($n = 7$) and quantification of osteoblast number per bone surface (Ob.N/BS). Scale bar: 100 μm. (O) Serum ELISA for P1NP ($n = 8$). Data are shown as mean ± SD. Two-tailed Student's unpaired t test (C, D, F). Two-tailed Student's paired t test (H, J, K, L, N, O). Source data are available online for this figure.

that many amino acid, purine and pyrimidine metabolic pathways were significantly reduced upon AOA treatment (Appendix Fig. S3C). Consistent with reduced amino acid and nucleotide abundance, AOA limited bone resorption in vitro similar to BPTES treatment (Fig. 7K; Appendix Fig. S3D). Together, these data demonstrate osteoclasts rely on glutamate transamination reactions downstream of GLS to support amino acid and nucleotide synthesis to regulate osteoclast differentiation and bone resorption.

### Inhibiting glutaminolysis protects mice from estrogen deficiency induced bone loss

Collectively, our data indicate that glutaminolysis is essential to support osteoclastogenesis and bone resorption. We questioned if we could exploit this metabolic vulnerability to limit bone resorption in conditions of estrogen deficiency. In vitro differentiation assays in the presence of the GLS inhibitor Telaglenastat (also known as CB-839) found that osteoclast differentiation is 200 times more sensitive to CB-839 than osteoblast differentiation (Appendix Fig. S4). This suggests that CB-839 can be used to preferentially inhibit osteoclast-mediated bone resorption without disrupting osteoblast-mediated bone formation. To test this, we performed ovariectomy (OVX) or sham surgery in 10-week-old female mice and orally treated with either CB-839 or vehicle (Fig. 8A). OVX resulted in robust bone loss in vehicle-treated mice, which was mitigated by CB-839 treatment (Fig. 8B–J). As expected, OVX increased osteoclast numbers and bone resorption as shown by increased serum CTX-1, which was prevented by CB-839 (Fig. 8K–P). Importantly, CB-839 did not affect osteoblast numbers or activity as shown by dynamic histomorphometry (Fig. 8Q–U). Consistent with the conclusion that CB-839 functions primarily in osteoclasts to prevent bone loss, genetic ablation of Gls in myeloid cells using LysmCre also prevented OVX-induced bone loss (Fig. EV5). Collectively, these data demonstrate that pharmacological inhibition of GLS and glutaminolysis may be a valuable antiresorptive strategy to block estrogen deficiency-induced bone loss.

## Discussion

Here we present evidence that glutaminolysis functions as a lynch pin to sustain metabolic reprogramming associated with osteoclastogenesis. Osteoclastogenesis is characterized by broad metabolic changes including increased abundance of amino acid and nucleotide metabolites and reduced abundance of metabolites in energetic pathways. This metabolic signature is governed by the increased consumption and metabolism of glutamine, an important precursor of both amino acids and nucleotides. Genetically ablating glutaminolysis in either myeloid lineage cells or osteoclast precursors inhibited osteoclast differentiation and bone resorption resulting in increased bone mass. Conversely, genetically increasing glutaminolysis in myeloid cells enhanced osteoclastogenesis and accelerated bone loss. Highlighting the therapeutic implications of these findings, inhibiting glutaminolysis using CB-839 prevented ovariectomy induced bone loss in mice. Collectively, our data provide strong evidence of the necessity of glutaminolysis to regulate osteoclast metabolism and promote osteoclast differentiation and bone resorption in mice.

Our findings are congruent with a recent study which demonstrated that metabolic changes limit osteoclast function (Rohatgi et al, 2023). For example, genetic ablation of the deubiquitinating enzyme BAP1 resulted in increased expression of the cysteine exchanger SLC7A11, increased GSH, reduced mitochondrial activity, and reduced abundance of several amino acids with no change in glutamine and increased glutamate and aspartate. These metabolic changes were attributed in part to altered glutamine flux away from the TCA cycle to GSH synthesis in BAP1 knockout myeloid cells (Rohatgi et al, 2023). Our data support this conclusion as reducing glutamine flux by inhibiting glutaminolysis prevented osteoclast differentiation and function. We do not think altering glutamine flux affects mitochondrial respiration directly, as others have demonstrated glutamine is not a primary oxidative substrate in osteoclasts (Kushwaha et al, 2022; Li et al, 2020). Moreover, GLS inhibition did not affect ATP levels (Fig. 2). Interestingly, inhibiting glutaminolysis specifically limited the induction of genes associated with osteoclast fusion (e.g., Dc-stamp and Atp6v0d2) and terminal differentiation (e.g., Catk) without affecting NFATC1 expression. This is peculiar as these genes are directly regulated by NFATC1 (Kim et al, 2008; Matsumoto et al, 2004). The effects of inhibiting glutaminolysis on osteoclast differentiation, fusion and function are likely multifaceted. We posit that reducing amino acid and nucleotide synthesis are the prime culprits. Nucleotides and amino acids are essential for RNA and protein synthesis which govern the production of osteoclast regulatory proteins and collagenolytic enzymes. Inhibiting either RNA or protein synthesis in mature osteoclasts reduced bone resorption in vitro (Hall et al, 1994). Amino acids may regulate the mTOR pathway or autophagic proteins which are essential for the fusion of secretory vesicles to form the ruffled border and resorb bone (DeSelm et al, 2011). Thus, inhibiting glutaminolysis results in a loss of amino acids and nucleotides that should broadly limit osteoclast differentiation and function.

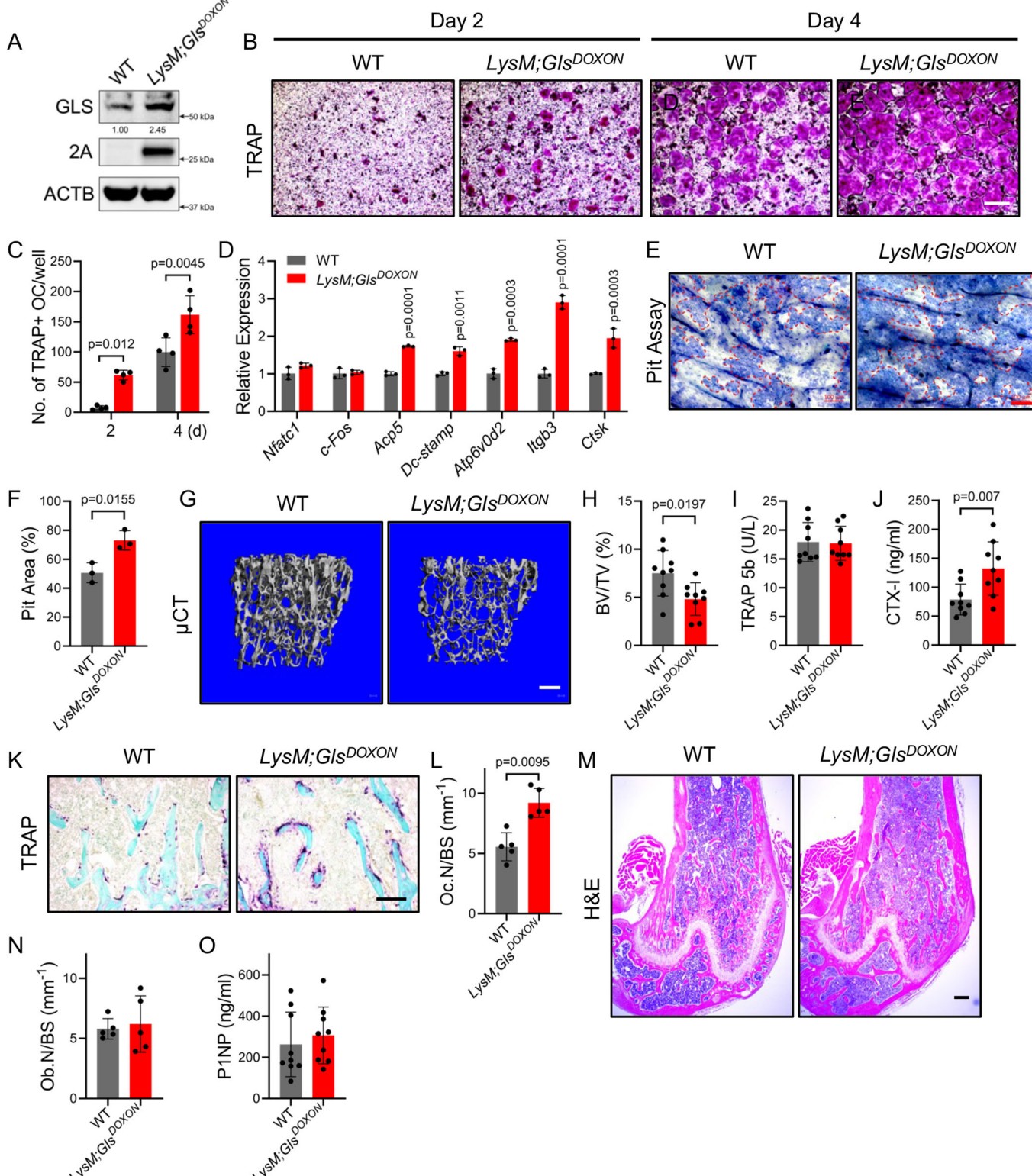

**Figure 5.  Increasing GLS in myeloid cells enhances osteoclastogenesis and bone resorption.**

(A) Western blot analyses of GLS expression in primary BMM derived from *LysMcre;Rosa*$^{rtTA}$ (WT) or *LysMcre;Gls*$^{DOXON}$ mice cultured with 100 ng/mL doxycycline for 48 h ($n = 3$). (B) TRAP staining of primary BMM isolated from WT or *LysMcre;Gls*$^{DOXON}$ female mice cultured with RANKL for 2 or 4 days in the presence of doxycycline ($n = 4$). Scale bar: 500 μm. (C) Quantification of TRAP-positive multinucleated mOC ($n = 4$ independent experiments). (D) qPCR showing enhanced expression of osteoclast marker genes in *LysM;Gls*$^{DOXON}$ BMM differentiated for 4 days in the presence of doxycycline ($n = 3$ independent experiments). (E, F) Representative toluidine blue staining (E) and quantification (F) showing enhanced resorption pit area in *LysM;Gls*$^{DOXON}$ BMM differentiated in the presence of doxycycline ($n = 3$ independent experiments). Scale bar: 100 μm. (G, H) Representative μCT (G) and calculated bone volume per tissue volume (BV/TV) (H) of trabecular bone in the distal femurs of 2-month-old WT and *LysM;Gls*$^{DOXON}$ female mice ($n = 9$ mice). Scale bar: 200 μm. (I–J) Serum ELISA for TRAP 5b (I) and CTX-I (J) from 2-month-old WT and *LysM;Gls*$^{DOXON}$ female mice ($n = 9$). (K–L) Representative TRAP staining (K) and quantification of osteoclast number per bone surface (Oc.N/BS) (L) from distal femurs from 2-month-old WT and *LysM;Gls*$^{DOXON}$ female mice ($n = 5$). Scale bar: 100 μm. (M, N) Representative H&E stain (M) and quantification of osteoblast number per bone surface (Ob.N/BS) (N) ($n = 5$). Scale bar: 100 μm. (O) Serum ELISA for P1NP ($n = 9$). Data are shown as mean ± SD. Two-tailed Student's unpaired $t$ test (C, D, F). Two-tailed Student's paired $t$ test (H, I, J, L, N, O). Source data are available online for this figure.

The finding that glutamate and aspartate are elevated in BAP1 knockout myeloid cells is intriguing (Rohatgi et al, 2023). Aspartate is a prominent product of glutaminolysis in osteoclasts (Figs. 2 and 7). This suggests that maintaining aspartate homeostasis is a priority in osteoclasts. Over 20% of aspartate was $M + 4$ labeled in mOC (Fig. 2H) indicating GLN-derived α-KG is converted to oxaloacetate which is a substrate for glutamate transamination to synthesize aspartate. Glutamate transamination couples the anabolic utilization of glutamine carbon and nitrogen. Inhibiting glutamate transamination broadly reduced amino acids and nucleotides similar to GLS inhibition. This suggests glutamate transamination allows osteoclasts to utilize glutamine in an economical way coupling amino acid and nucleotide biosynthesis with α-KG production. αKG metabolism is compartmentalized in the cytoplasm and mitochondria (Parker et al, 2021). Cytoplasmic αKG is a cofactor for dioxygenases that mediate histone and DNA demethylation whereas mitochondrial αKG is a substrate for transamination and contributes to energetics through the TCA cycle (Rose et al, 2011). Two recent studies demonstrated that αKG exerts epigenetic regulation of osteoclastogenesis (Lee et al, 2021; Stegen et al, 2024). For example, PSAT1-dependent serine synthesis provides αKG to regulate *Nfatc1* expression and osteoclast differentiation. Our data indicates PSAT1 functions independent of glutaminolysis as we observed negligible enrichment of glutamine nitrogen in serine (Fig. 7H) and GLS inhibition did not affect *Nfatc1* or *cFos* upregulation (Fig. 3C,D). This suggests that glutamate metabolism is also compartmentalized in osteoclasts with glutamine providing mitochondrial glutamate for mitochondrial transaminases like GOT2. Cytoplasmic transaminases like PSAT1 likely utilize a separate source of glutamate that is glutamine-independent. Moreover, we think a role for glutamine-derived αKG in cellular energetics is unlikely as inhibiting GLS does not affect ATP levels despite reducing αKG abundance (Kushwaha et al, 2022; Li et al, 2020). Interestingly, other groups found dimethyl-αKG (DMαKG) supplementation could replace glutamine during osteoclastogenesis in vitro (Indo et al, 2013; Peng et al, 2024). In contrast, we found DMαKG supplementation was unable to rescue osteoclast differentiation in the absence of glutamine (Fig. EV4A–H). The reason for this discrepancy is not immediately clear. In pancreatic cancer cells, DMαKG is imported into the mitochondria where it contributes to glutamate and increases aspartate abundance (Parker et al, 2021). Thus, DMαKG may alleviate some aspects of glutamine deficiency by increasing glutamate and aspartate to enable amino acid and nucleotide synthesis.

Outside of PSAT1, there is little known about the requirements for individual amino acid transaminases in osteoclasts. A recent study concluded BCAT1 was important for osteoclast maturation by providing branched-chain amino acids (Go et al, 2022). While our RNA-seq data indicates *Bcat1* is lowly expressed, glutamine nitrogen incorporation into the BCAAs was increased during differentiation consistent with a role for BCAT activity in osteoclasts (Fig. 7E–G). Given the relatively low labeling of the BCAAs compared to aspartate, we conclude other transaminases, like the glutamate oxaloacetate transaminases (GOTs) that generate aspartate, function downstream of GLS in osteoclasts. While unstudied in osteoclasts, GOT activity is essential for many metabolic processes that are essential for osteoclastogenesis, including nucleotide metabolism, glycolysis, the TCA cycle, REDOX homeostasis, and electron transport chain activity.

Osteoclastogenesis and bone resorption are energy-intensive processes that require robust ATP synthesis. For example, osteoclasts are mobile and continuously produce and secrete protons and collagenolytic enzymes to remove mineral and bone matrix, respectively (Arnett and Orriss, 2018). Consistent with increased energetic demands, mitochondrial biogenesis, oxygen consumption, and electron transport chain components increase significantly during osteoclastogenesis. Moreover, mature osteoclasts have abundant active mitochondria (Arnett and Orriss, 2018; Da et al, 2021; Indo et al, 2013; Lemma et al, 2016; Zeng et al, 2015); (Ishii et al, 2009; Zhang et al, 2018). Proteomic studies in RAW 264.7 cells found that osteoclasts prioritize ATP for bone resorption by reducing proteins involved in basic cellular functions (e.g., gene expression and protein synthesis) (An et al, 2014). It is intriguing then those metabolites associated with energetic pathways like glycolysis, mitochondrial ETC and β-oxidation were enriched in BMM and reduced in pOC and mOC. It is also important to note the abundance of intracellular ATP was reduced in both pOC and mOC (Figs. 1D and EV1J) which is consistent with a previous report (Miyazaki et al, 2012). We interpret these data to indicate that flux through these energetic pathways increases in pOC and mOC resulting in reduced abundance of the respective intracellular metabolites. This would be consistent with recent work from the Long and Riddle labs showing both glycolysis and oxidative phosphorylation as well as glucose uptake and oleate oxidation increase substantially during osteoclastogenesis (Kushwaha et al, 2022; Li et al, 2020; Song et al, 2023). While blocking OXPHOS inhibits osteoclastogenesis, it is important to note that a major function of mitochondrial activity is to support aspartate biosynthesis and subsequent nucleotide biosynthesis (Birsoy et al, 2015; Sullivan et al, 2015). It is not clear if the reported effects of OXPHOS inhibition on osteoclastogenesis were due to changes in energy availability or aspartate and nucleotide

**Figure 6. Amino acids and nucleotides can rescue osteoclast differentiation and bone resorption in the absence of glutaminolysis.**

The effect of supplementing non-essential amino acids (NEAAs) and/or nucleotides on osteoclast differentiation, fusion and bone resorption as measured by TRAP staining (**A**, scale bar: 500 μm), phalloidin staining (**B**, scale bar: 100 μm), pit assay (**C**, scale bar: 100 μm) or qPCR analysis of *Ctsk* mRNA expression ($n = 3$) (**G**). (**D**) Quantification of TRAP-positive multi-nuclei cells from (**A**) ($n = 5$ independent experiments). (**E**) Quantification of the number of actin belts from (**B**) ($n = 4$). (**F**) Quantification of the resorption pit area from (**C**) ($n = 4$). Data are shown as mean ± SD. Two-way ANOVA (**D–G**). Source data are available online for this figure.

biosynthesis or some combination therein. Given this uncertainty, future studies into the role of mitochondria and nucleotide biosynthesis in osteoclasts are warranted.

In summary, the work presented here revealed that glutaminolysis is a metabolic lynch pin in osteoclasts and is essential for osteoclast differentiation and bone resorption. Moreover, osteoclasts are reminiscent of cancer cells which have increased reliance on glutamine. Targeting glutaminolysis using CB-839 is currently being developed as a potential cancer treatment (Harding et al, 2021). Remarkably, CB-839 treatment prevented ovariectomy induced bone loss in mice, highlighting the clinical potential of CB-839 outside of cancer. Importantly, the CB-839 data is consistent with several independent studies showing that pharmacological inhibition of either glutamine uptake or glutaminolysis prevented osteoclast expansion and bone loss in ovariectomized mice (Guo et al, 2024; Peng et al, 2024; Shen et al, 2018). To our knowledge, the effect of CB-839 treatment on bone mass in humans has not been reported. Regardless, our data support the notion that drugs designed to target cancer metabolism can be repurposed to limit excessive bone resorption. Furthermore, our data suggests that like PHGDH (Stegen et al, 2024), targeting one or more mitochondrial glutamate-dependent transaminases might also be worth exploring as a therapeutic option to limit excessive or pathological bone resorption.

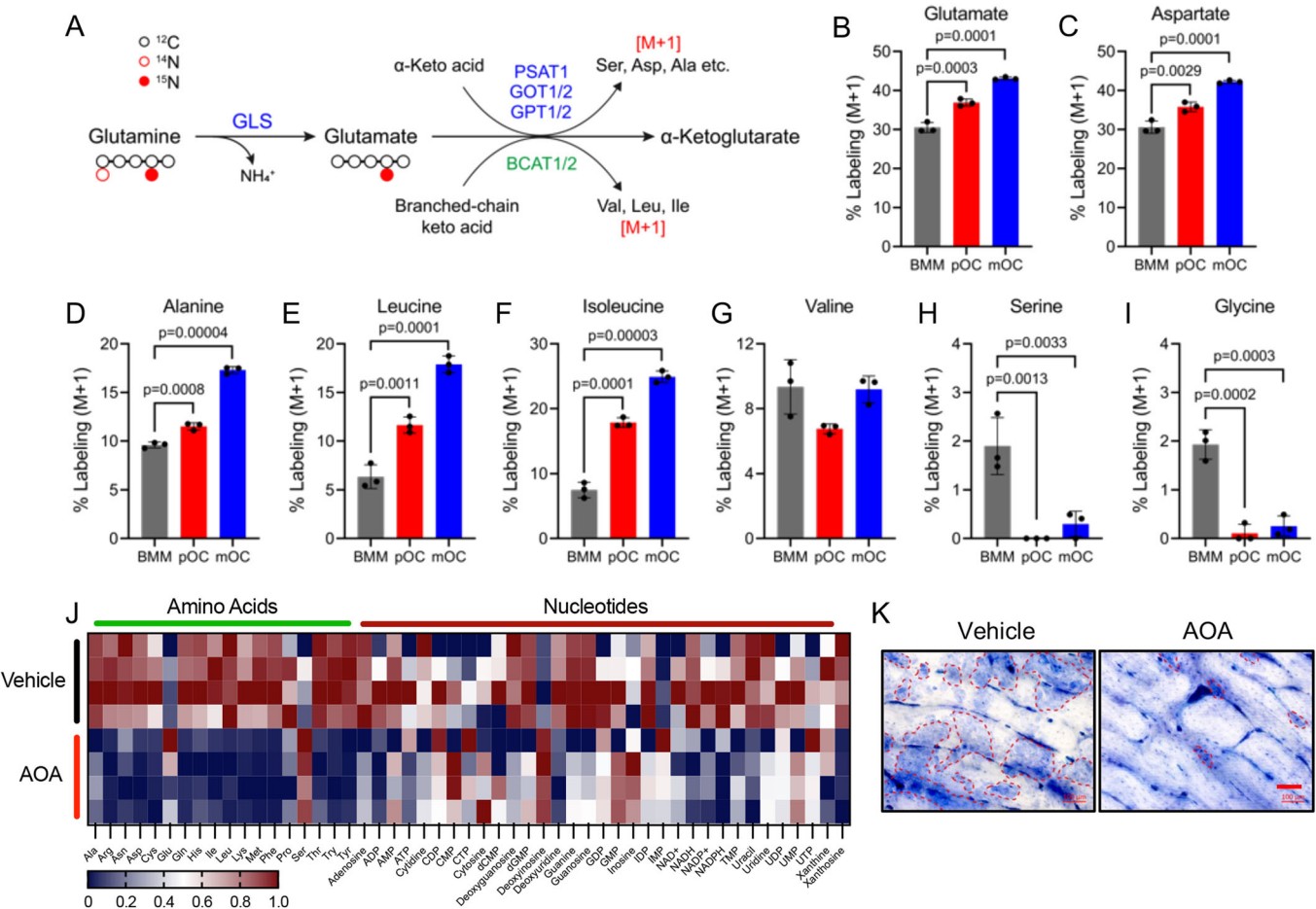

**Figure 7. Glutamate transamination is essential to maintain amino acid and nucleotide abundance.**

(A) Schematic depicting glutamine nitrogen metabolism and isotope tracing strategy. (B–I) Graphical depiction of percent labeling of glutamate (B), aspartate (C), alanine (D), leucine (E), isoleucine (F), valine (G), serine (H), and glycine (I) from [α-$^{15}$N]-glutamine in BMM, pOC, and mOC ($n = 3$ independent experiments). (J) Heatmap showing relative abundance of metabolites in 200 μM AOA- or vehicle-treated mOC as determined by LC-MS/MS ($n = 4$). (K) Representative images of resorption pit staining ($n = 3$). The calculated resorbing area is in Appendix Fig. S3D. AOA (200 μM) was added on day 4-6 of the culture period. Scale bar: 100 μm. Data are shown as mean ± SD. One-way ANOVA (B–I). Source data are available online for this figure.

# Methods

### Reagents and tools table

| Reagent/resource | Reference or source | Identifier or catalog number |
|---|---|---|
| **Experimental models: organisms/strains** | | |
| Mouse: *Gls^fl/fl: Gls^tm2.1Sray/J* | The Jackson Laboratory | RRID: IMSR_JAX:017894 |
| Mouse: *Lyz2^tm1(cre)Ifo/J* | The Jackson Laboratory | RRID: IMSR_JAX:004781 |
| Mouse: *Tg(Csf1r-cre/Esr1*)1Jwp/J* | The Jackson Laboratory | RRID: IMSR_JAX:019098 |
| Mouse: *Rosa26^LSL-rtTA3* | The Jackson Laboratory | RRID: IMSR_JAX:029617 |
| Mouse: *C57Bl/6 J* | The Jackson Laboratory | RRID: IMSR_JAX:000664 |
| Mouse: *tetO-Gls-t2a-mCherry* | This paper | N/A |

| Reagent/resource | Reference or source | Identifier or catalog number |
|---|---|---|
| **Recombinant DNA** | | |
| pMM400sfi vector | Addgene http://n2t.net/addgene:34925 | Cat# 34925; RRID: Addgene_34925 |
| **Antibodies** | | |
| Anti-GLS | Cell Signaling Technology | Cat# 56750 |
| Anti-NFATc1 | BD Biosciences | Cat# 556602; RRID: AB_396478 |
| Anti-GOT1 | Proteintech | Cat# 14886-1-AP; RRID: AB_2113630 |
| Anti-GOT2 | Proteintech | Cat# 14800-1-AP; RRID: AB_2247898 |
| Anti-GPT2 | Proteintech | Cat# 16757-1-AP; RRID: AB_2112098 |

| Reagent/resource | Reference or source | Identifier or catalog number |
|---|---|---|
| Anti-SLC1A5 | Cell Signaling Technology | Cat# 5345; RRID: AB_10621427 |
| Anti-β-actin | Cell Signaling Technology | Cat# 4970; RRID: AB_2223172 |
| Anti-mCherry | Abcam | Cat# ab167453; RRID: AB_2571870 |
| HRP-linked anti-rabbit IgG | Cell Signaling Technology | Cat# 7074; RRID: AB_2099233 |
| HRP-linked anti-mouse IgG | Cell Signaling Technology | Cat# 7076; RRID: AB_330924 |
| Alexa fluor 633 anti-rabbit IgG | ThermoFisher | Cat# A21070; RRID: AB_2535731 |
| **Oligonucleotides and other sequence-based reagents** | | |
| See Appendix Table S1 for qPCR primers | https://www.idtdna.com/ | N/A |
| **Chemicals, enzymes and other reagents** | | |
| Ascorbic acid | Sigma | Cat# A4544 |
| β-glycerophosphate | Sigma | Cat# G9422 |
| L-glutamine | Sigma | Cat# G7513 |
| BPTES | Sigma | Cat# SML0601 |
| AOA | Sigma | Cat# C13408 |
| CB-839 | MedChemExpress | Cat# HY-12248 |
| HP-β-CD | MedChemExpress | Cat# HY-101103 |
| MEM Non-Essential Amino Acids Solution (100X) | ThermoFisher | Cat# 11140050 |
| EmbryoMax Nucleotides (100X) | Sigma | Cat# ES-008-D |
| (U-$^{13}$C)-Glutamine | Sigma | Cat# 605166 |
| (α-$^{15}$N)-Glutamine | Sigma | Cat# 486809 |
| L-(2,3,4-$^3$H)-Glutamine | Perkin Elmer | Cat# NET551250UC |
| L-[3,4-$^3$H]-Glutamic acid | Perkin Elmer | Cat# NET490250UC |
| L-[1,2-$^{14}$C]-Alanine | Perkin Elmer | Cat# NET856250UC |
| L-[2,3,4,5-$^3$H]-Proline | Perkin Elmer | Cat# NET483250UC |
| M-CSF | R&D Systems | Cat# 216-MC |
| RANKL | R&D Systems | Cat# 462-TEC |
| Proteinase K | Sigma | Cat# 1.24568 |
| TRIzol | ThermoFisher | Cat# 15596018 |
| Triton X-100 | Sigma | Cat# T8787 |
| Doxycycline | Sigma | Cat# D9891 |
| Tamoxifen | Sigma | Cat# T5648 |
| 25% Glutaraldehyde solution | Sigma | Cat# G6257 |
| Toluidine blue | Sigma | Cat# 89640 |
| Phalloidin-iFluor 488 | Abcam | Cat# ab176753 |
| Sodium borate decahydrate | Sigma | Cat# SX0355 |
| Sodium citrate | Sigma | Cat# W302600 |
| Alizarin red S | Sigma | Cat# A5533 |

| Reagent/resource | Reference or source | Identifier or catalog number |
|---|---|---|
| Calcein | Sigma | Cat# C0875 |
| Trypsin-EDTA | ThermoFisher | Cat# 25200-072 |
| Collagenase P | Roche | Cat# 11213873001 |
| Fetal bovine serum | ThermoFisher | Cat# 16000-044 |
| DNase I | ThermoFisher | Cat# 18068015 |
| SYBR Green | Bio-Rad | Cat# 1725275 |
| Protease inhibitor cocktail | Roche | Cat# 11697498001 |
| Phosphatase inhibitor | Roche | Cat# 04906837001 |
| Mounting medium with DAPI | Vector Laboratories | Cat# H-2000 |
| Pen Strep | Gibco | Cat# 15140-122 |
| Red blood cell lysis buffer | Roche | Cat# 57350100 |
| Clarity ECL substrate | Bio-Rad | Cat# 1705060 |
| Super Signal West Femto substrate | ThermoFisher | Cat# PI34095 |
| One-step NBT/BCIP solution | ThermoFisher | Cat# PI34042 |
| iScript cDNA Synthesis Kit | Bio-Rad | Cat# 1708841 |
| Acid Phosphatase (TRAP) Kit | Sigma | Cat# 387A |
| Mouse P1NP ELISA Kit | Immunodiagnostic Systems | Cat# AC-33F1 |
| Mouse CTX-I ELISA Kit | Immunodiagnostic Systems | Cat# AC-02F1 |
| Mouse TRAP5b ELISA Kit | Immunodiagnostic Systems | Cat# SB-TR103 |
| **Software** | | |
| Image J | https://imagej.nih.gov/ij/ | N/A |
| Graphpad Prism 6 | https://www.graphpad.com/ | N/A |
| **Other** | | |
| Bone slices | Immunodiagnostic Systems | Cat# DT-1BON1000-96 |
| α-MEM | ThermoFisher | Cat# 12561-056 |
| Glutamine Free α-MEM | Corning | Cat# 15-012-cv |
| PicoLab Rodent Diet 290 | LabDiet | Cat# 5053 |
| Doxycycline Diet | ENVIGO | Cat# TD.120769 |

## Methods and protocols

### Mouse strains

*Gls^flox^* mouse strain is as previously described (Yu et al, 2019). *C57Bl/6J*, *Rosa26^LSL-rtTA3^*, *LysMCre* and *Csfr1Cre^ERT2^* strains were obtained from the Jackson Laboratory. The *tetO-Gls-t2a-mCherry* transgenic mice were generated by cyagen who microinjected the pMM400sfi vector containing a codon optimized *Gls-t2a-mCherry* cassette. pMM400Sfi was a gift from Mark Mayford (Addgene plasmid # 34925; http://n2t.net/addgene:34925; RRID: Addgene_34925). Pups were screened by PCR using the following primers: F1

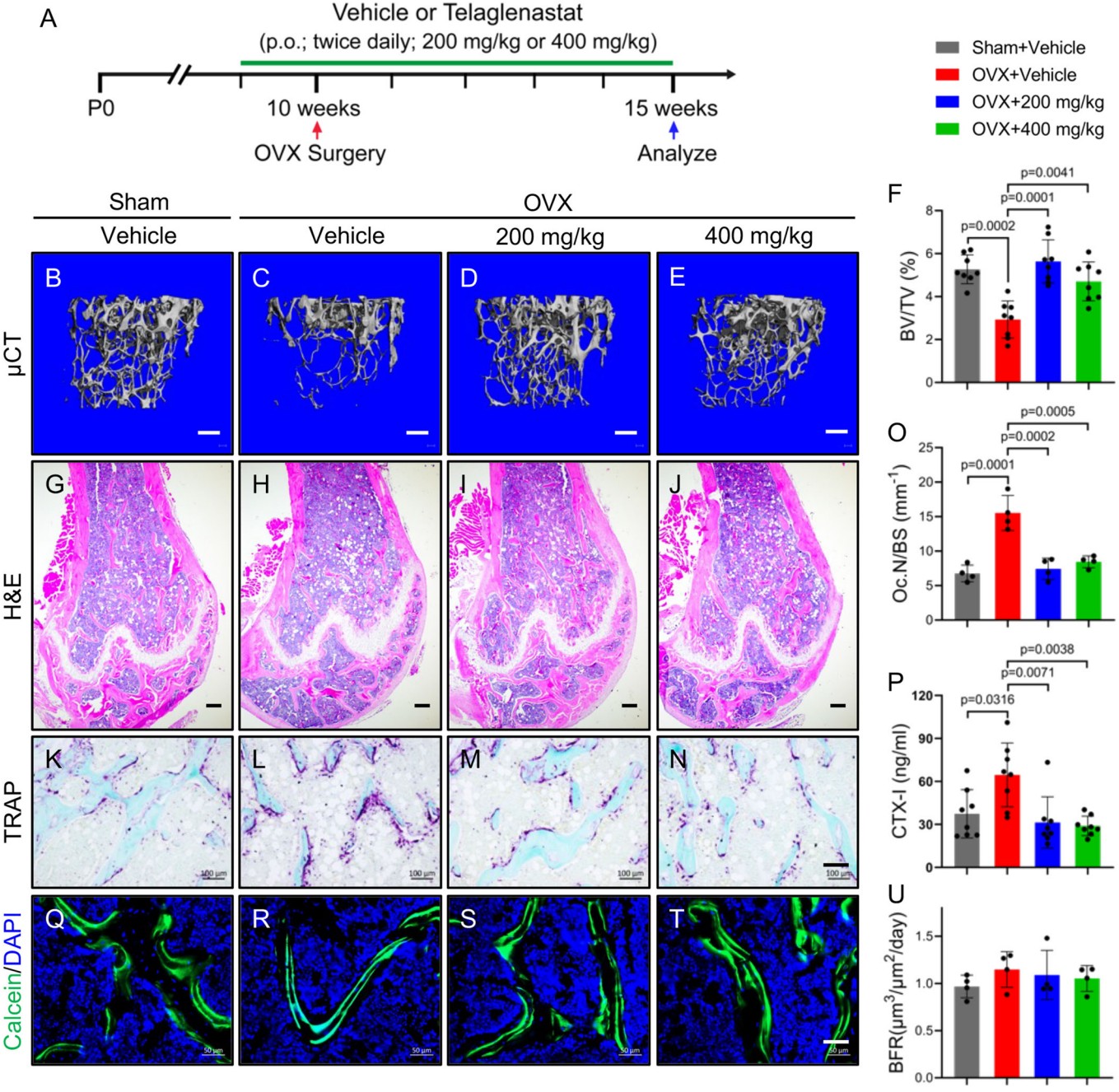

**Figure 8. Treatment with the GLS inhibitor CB-839 prevents OVX-induced bone loss.**

(A) Schematic showing the experimental design. Representative μCT images (**B–F**, scale bar: 200 μm), H&E staining (**G–J**, scale bar: 100 μm), TRAP staining (**K–O**, scale bar: 100 μm), Serum CTX-1 (**P**) and double calcein labeling (**Q–U**, scale bar: 50 μm) of distal femur trabecular bone of sham and OVX mice treated with vehicle or CB-839 (n = 8 mice). (F) The calculated BV/TV from μCT images (n = 8 mice). (O) Osteoclast number per bone surface (Oc.N/BS) quantified from TRAP stains (n = 4 mice). (P) Serum ELISA of CTX-I (n = 8 mice). (U) Bone formation rate (BFR) calculated from the double calcein labeling (n = 4 mice). Data are shown as mean ± SD. Two-way ANOVA (**F, O, P, U**). Source data are available online for this figure.

TTGACCTCCATAGAAGACACCGG, R1 AAGAGCAGCTCCCGT AGCATC, 60 °C annealing temperature, expected band size 437 bp. Positive pups were confirmed by PCR using the following primers: F2 GCATGGACGAGCTGTACAAGTAG, R2 TGCTCCCATTCAT-CAGTTCCA, 60 °C annealing temperature, expected band size 259 bp. Multiple founders were established and characterized which

will be described elsewhere. All mice were backcrossed onto the *C57Bl/6 J* background for five generations prior to experiments. The littermate *LysMCre;Rosa26^{LSL-rtTA}* mice were used as controls. GLS expression was induced by feeding chow containing 0.625 g doxycycline hyclate/kg beginning at 1 month of age and bone phenotypes were evaluated at 2 months of age. *Csfr1Cre^{ERT2};Gls^{fl/fl}* mice were given

tamoxifen (75 mg/kg body weight) via daily intraperitoneal injection for a total of 5 consecutive days at 1 month of age and bone parameters were evaluated at 2 months of age. All mice were housed at 23 °C on a 12-h light/dark cycle with free access to water and PicoLab Rodent Diet 290. All animal use was approved by the Institutional Animal Care and Use Committee (IACUC) at the University of Texas Southwestern Medical Center at Dallas, Texas.

### Micro-computed tomography (µCT)

Micro-computed tomography (µCT) (µCT45, Scanco Medical AG, set at 55 kVp and 145 µA, Voxel size: 4.9 µm) was used for three-dimensional reconstruction and quantification of bone parameters. For quantification of bone mass in the long bone, femurs were isolated, fixed, immobilized, and scanned. Bone parameters were quantified from 400 slices directly underneath the growth plate with the threshold set at 300. For all animal studies, both male and female mice were analyzed. The assessment of all animals was performed in a blinded and coded manner.

### Ovariectomy surgery and CB-839 treatment

For ovariectomy experiment, bilateral OVX or sham operation was performed on 10-week-old female mice. The mice were sacrificed 5 weeks after surgery, uterine atrophy was first confirmed and then bones were collected for µCT and histological analysis. For CB-839 treatment, mice were administered either 200 or 400 mg/kg CB-839, or vehicle (25% (w/v) hydroxypropyl-β-cyclodextrin (HP-β-CD) in 10 mmol/L citrate (pH 2.0)) by oral gavage twice daily. CB-839 was formulated as a solution at 20 mg/ml (w/v). The dose volume for all groups was 10 ml/kg.

### Histomorphometry and immunostaining

Bone static histomorphometry was performed on femurs fixed in 10% buffered formalin for 48 h at 4 °C followed by 2-week decalcification in 14% EDTA. Decalcified bones were embedded in paraffin and sectioned at 5 µm thickness. Hematoxylin and eosin (H&E) and tartrate-resistant acid phosphatase (TRAP) staining were performed following standard protocols. For mCherry immunostaining, antigen retrieval was performed by incubating tissue sections in 10 µg/mL proteinase K for 10 min followed by incubation in 3% $H_2O_2$ (v/v in water) for 10 min to block endogenous peroxidase activity. Subsequently, sections were incubated overnight at 4 °C with primary antibody against mCherry and then incubated with Fluor 633-conjugated anti-rabbit IgG at room temperature for 30 min and mounted with antifade mounting medium with DAPI. Static histomorphometry was quantified using Image J (https://imagej.nih.gov/ij/). All Image analysis and quantifications were performed in a blinded and coded manner.

### Serum analysis

Mouse blood samples were collected via cardiac puncture after 3 h of fasting. Blood samples were allowed to clot for 2 h at room temperature before centrifugation for 10 min at approximately 1000×g. Freshly prepared serum samples were stored at −80 °C for later analyses. Serum P1NP, CTX-I, and TRAP5b levels were measured by using P1NP ELISA kit, CTX-I ELISA kit, and TRAP5b ELISA kit respectively according to the manufacturer's instructions.

### Calcein double labeling

Calcein (20 mg/kg) was intraperitoneally injected at 7 and 2 days prior to sacrifice, respectively. Freshly isolated bones were fixed in

4% paraformaldehyde overnight and dehydrated in 30% sucrose for 24 h and embedded in OCT for frozen sections using CryoJane Tape-Transfer System (Electron Microscopy Sciences, Hatfield, PA, USA). Dynamic histomorphometry were quantified using ImageJ.

### Monocyte/macrophage isolation and osteoclast culture

Primary bone marrow macrophages (BMMs) were isolated from age and sex-matched mutant and control mice. Briefly, the femur and tibias were isolated, and all connective tissue was removed. The epiphyses were removed, and marrow was collected by brief centrifugation. Red blood cells were lysed by red blood cell lysis buffer for 1 min. Then bone marrow cells were cultured in α-MEM media containing 10% FBS, 100 U/mL penicillin G, 100 µg/mL streptomycin, and 25 ng/mL M-CSF under a humidified atmosphere of 5% $CO_2$ at 37 °C. After 2 days, cells were harvested, counted, seeded at a density of $4.5 \times 10^4/cm^2$, and cultured in α-MEM media supplemented with 25 ng/mL M-CSF and 40 ng/mL RANKL for another 4–5 days. Culture media were exchanged every other day. BPTES (10 µM), AOA (200 µM) or vehicle (DMSO) was included in the culture media as indicated in the figure legend. TRAP staining was performed with an acid phosphatase leukocyte diagnostic kit in accordance with the manufacturer's instructions. To visualize actin belts, the osteoclasts were fixed with 4% formaldehyde for 10 mins and permeabilized with 0.1% Triton X-100 for 3 mins and then stained with Phalloidin-iFluor 488 at room temperature for 1 h. For rescue experiments, 1×MEM non-essential amino acids (NEAAs) solution and/or 1× EmbryoMax nucleoside mixture were supplemented from the beginning (day 0) or at a later phase (day 4) of the osteoclast culture as indicated in the figure legends. To evaluate bone resorption, bone slices were labeled, placed in a 96-well plate with labeled side down and sterilized in a sterile hood with UV light the day before the experiment. On the day of the experiment, bone marrow derived macrophages were plated at 10,000 cells/well and cultured in α-MEM media supplemented with 25 ng/mL M-CSF and 40 ng/mL RANKL for 6 days. Culture media were exchanged every other day. Bone slices were then fixed with 2.5% glutaraldehyde in PBS for 30 min at room temperature. Adherent cells were removed from the bone slice by sonicating in distilled water at 50–60 Hz for 5 min. Bone slices were then completely dried on Whatman filter paper and stained with 1% toluidine blue (w/v in 1% sodium borate) for 4 min at room temperature. The visualized pit area was quantified in 5 random sites using Image J.

### GC-MS analysis of glutaminolysis

BMMs were cultured with M-CSF alone or together with RANKL for 2–4 days prior to incubation in 2 mM [U-$^{13}$C]-glutamine or [α-$^{15}$N]-glutamine for 6 h. At the end of the incubation, cells were washed three times with ice-cold normal saline solution and extracted twice with 0.5 mL 80% (vol/vol) methanol (cooled to −80 °C) on dry ice. The methanol extract was subjected to 3 freeze-thaw cycles between liquid nitrogen and 37 °C and then centrifuged at ~20,160×g for 15 min in a refrigerated centrifuge after vortex. The supernatant was completely dried by $N_2$ gas. The dried residues were resuspended in 25 µL of methoxyamine hydrochloride (2% (w/v) in pyridine) and incubated at 40 °C for 90 min on a heating block. After brief centrifugation, 35 µL of MTBSTFA + 1% TBDMS was added, and the samples were incubated at 60 °C for 30 min. The derivatized samples were centrifuged for 5 min at ~20,160×g, and the supernatants were transferred to GC vials

for GC-MS analysis. The injection volume was 1 μL, and samples were injected in split or splitless mode depending on analyte of interest. GC oven temperature was held at 80 °C for 2 min, increased to 280 °C at 7 °C/min, and held at 280 °C for a total run time of 40 min. GC-MS analysis was performed on an Agilent 7890B GC system equipped with a HP-5MS capillary column (30 m, 0.25 mm i.d., 0.25 μm-phase thickness; Agilent J&W Scientific), connected to an Agilent 5977 A Mass Spectrometer operating under ionization by electron impact at 70 eV. Helium flow was maintained at 1 mL/min. The source temperature was maintained at 230 °C, the MS quad temperature at 150 °C, the interface temperature at 280 °C, and the inlet temperature at 250 °C. Mass spectra were recorded in selected ion monitoring (SIM) mode with 4 ms dwell time.

### Untargeted metabolomic analysis by LC-MS/MS

At the end of culture, cells were washed three times with ice-cold normal saline solution and extracted twice with 0.5 mL 80% (vol/vol) methanol (cooled to −80 °C) on dry ice. The methanol extract was subjected to 3 freeze-thaw cycles between liquid nitrogen and 37 °C and then centrifuged at ~20,160×$g$ at 4 °C for 15 min after vortex. The supernatant was completely dried by $N_2$ gas. The dried residues were redissolved in 50 μL milli-Q water and transferred to LC vials for LC-MS analysis. All metabolomic analyses were performed using a Millipore Sigma ZIC-pHILIC column (2.1 × 150, 5 μm) with a binary solvent gradient. Mobile phase A was water containing 10 mM ammonium acetate, pH 9.8 with ammonium hydroxide; mobile phase B was 100% acetonitrile. Gradient separation proceeded as follows: from 0 to 15 min mobile phase B was ramped linearly from 90% to 30%; from 15 min to 18 min, mobile phase B was held at 30%; from 18 min to 19 min, mobile phase B was ramped linearly from 30% to 90%; mobile phase B was held at 90% from 19 min to 27 min to regenerate the initial chromatographic environment. Throughout, the solvent flow rate was kept at 250 μL/min and the column temperature was maintained at 25 °C. For low abundance samples, 20 μL of sample was injected onto the column. For high abundance samples, 10 μL was injected. Mass spectrometry data were acquired using a Thermo Scientific (Bremen, Germany) QExactive HF-X or Orbitrap Fusion Lumos mass spectrometer (LC-MS/MS). A polarity-switching MS1 only acquisition method was used. Each polarity was acquired at a resolving power of 120,000 full width at half maximum (FWHM); the automatic gain control (AGC) target was set to 1,000,000 with a maximum inject time of 50 ms. The scan range was set to 80–1200 Daltons. Results were normalized with DNA contents.

### Amino acid uptake assay

Cells were washed once with PBS, followed by two additional washes with Krebs Ringers HEPES buffer (KRH) (120 mM NaCl, 5 mM KCl, 2 mM CaCl$_2$, 1 mM MgCl$_2$, 25 mM NaHCO$_3$, 5 mM HEPES, 1 mM D-glucose, pH=8.0). Cells were then incubated for 5 min in KRH containing 4 μCi/mL of either L-[2,3,4-³H]-glutamine, L-[3,4-³H]-glutamic acid, L-[1,2-¹⁴C]-alanine, or L-[2,3,4,5-³H]-proline. The consumption was terminated by 3 washes with ice-cold KRH. Cells were then lysed with 1 mL 1% SDS and mixed with 8 mL scintillation cocktail. Radioactivity was measured using a Beckman LS6500 scintillation counter. Results were normalized with DNA contents.

### Glutaminase activity assay

Cells were washed three times with Hanks Buffered Saline Solution (HBSS) and cultured for 20 min in α-MEM media containing 2 mM unlabeled glutamine and 4 μCi/mL L-[2,3,4-³H]-glutamine. Glutaminase activity was terminated by three washes with ice-cold HBSS. Cells were scraped into 1 mL ice-cold milli-Q water and sonicated for 1 min with 1 s pulses at 20% amplitude. After clarification, cell lysates were bound onto AG 1-X8 polyprep anion exchange columns. Uncharged glutamine was eluted with 2 mL H$_2$O for three times. Glutamate and downstream metabolites were eluted with 2 mL 0.1 M HCl for 3 times. Eluent fractions were pooled and combined with 4 mL scintillation cocktail and radioactivity was measured using a Beckman LS6500 Scintillation counter. Glutaminase activity was calculated using the following equation: radioactivity$_{HCl}$/(radioactivity$_{HCl}$ + radioactivity$_{H2O}$). Results were normalized with DNA contents.

### RNA isolation and qPCR

Total RNA from cells was extracted using TRIzol reagent following a standard RNA extraction protocol. First-strand cDNA was synthesized from 500 ng of total RNA with the iScript cDNA synthesis kit. qPCR was performed in technical and biological triplicates in a 96-well format on an ABI Quant Studio 3, using SYBR green chemistry. The PCR programs were 95 °C for 3 min followed by 40 cycles of 95 °C for 10 s and 60 °C for 30 s. Gene expression was normalized to *Actb* mRNA and relative expression was calculated using the $2^{-(\Delta\Delta Ct)}$ method. Primers were used at 0.1 μM, and their sequences are listed in Appendix Table S1.

### RNA-seq and analysis

Total RNA was extracted using TRIzol reagent following a standard RNA extraction protocol. 10 μg RNA sample was then treated with amplification grade DNase according to the manufacturer's instructions. NEBNext Ultra II RNA Library Prep Kit for Illumina (NEB) was used to purify poly-A⁺ transcripts and generate libraries with multiplexed barcode adaptors. All samples passed quality control analysis using a Bioanalyzer 2100 (Agilent). High-throughput sequencing (100 bp, single end) was performed using the Illumina Hiseq 4000 in the UTSW McDermott Center Next Generation Sequencing Core Facility with a sequencing depth between 45 and 55 million reads per sample. Fastq files were quality checked using fastqc (v0.11.2). Reads from each sample were mapped to the mouse genome (mm10) using STAR (v2.5.3a). Read counts were generated using featureCounts and the differential expression analysis was performed using edgeR to estimate the transcript abundances as counts per million (CPM) values and calculate adjusted *P* value. Benjamini–Hochberg false discovery rate (FDR) procedure was used to correct for multiple testing. Differentially expressed genes (DEG) with FDR < 0.01 and absolute value of Log$_2$FC > 1 were identified and Gene Ontology analysis was performed using Panther (Mi et al, 2019). Single cell analysis was conducted on previously published data (Tsukasaki et al, 2020, Data Ref: Tsukasaki M, 2020) and analyzed using the *R* package *Seurat*.

### Western blotting

Cells were lysed in RIPA buffer containing 50 mM Tris–HCl (pH 7.4), 150 mM NaCl, 1% NP-40, 0.5% sodium deoxycholate, and

0.1% SDS supplemented with protease inhibitor cocktail and phosphatase inhibitors. Protein fractions were collected by centrifugation at ~15,000×$g$ at 4 °C for 10 min. Protein concentration was estimated by the BCA method. Protein samples were then mixed with 4× loading buffer and boiled for 10 min. 25 μg of total protein was loaded onto 4%–15% polyacrylamide gel, and then transferred onto polyvinylidene difluoride (PVDF) membrane. The membranes were blocked for 1 h at room temperature with 5% milk in TBST (TBS, 0.1% Tween 20) and then incubated with specific primary antibodies overnight at 4 °C. The following primary antibodies were used in this study: anti-SLC1A5, anti-GLS, anti-GOT1, anti-GOT2, anti-GPT2, anti-NFATc1, and anti-β-actin. All primary antibodies were diluted 1:1000 in TBST containing 5% milk. Membranes were then washed three times using TBST and further incubated with HRP-linked secondary antibody in 5% milk for 1 h at room temperature. All blots were developed using either the Clarity ECL substrate or the Super Signal West Femto substrate. Each experiment was repeated at least three times.

### Statistical analysis

Statistical analysis was performed using Graphpad Prism 10 software (https://www.graphpad.com/). All data are represented as mean values ± standard deviation. In cell culture studies, statistical significance was determined by an unpaired two-tailed Student's $t$ test. For animal studies, statistical significance was determined by a paired two-tailed Student's $t$ test comparing paired littermate controls, one-way ANOVA, or two-way ANOVA as indicated in the text. A $P$ value < 0.05 is considered statistically significant. All experiments were performed with $n \geq 3$ biological replicates.

## Data availability

The datasets produced in this study are available in the following databases: RNA-Seq data: Texas Data Repository https://doi.org/10.18738/T8/X6CDEQ. Metabolomics data: Texas Data Repository https://doi.org/10.18738/T8/X6CDEQ.

The source data of this paper are collected in the following database record: biostudies:S-SCDT-10_1038-S44319-024-00255-x.

## Peer review information

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

## Acknowledgements

We thank Sierra Root, Vishal Patel and Thomas J Carroll for helpful discussions; the CRI metabolomics Core for help with metabolomics and data

analysis, the McDermott Center Sequencing Core for help with mRNA-seq and data analysis. This work was supported by the National Institute of Arthritis and Musculoskeletal and Skin Diseases (NIAMS) grants (AR076325 and AR071967) to CMK The funders had no role in study design, data acquisition, decision to publish, or preparation of the manuscript.

## Author contributions

**Guoli Hu**: Formal analysis; Investigation; Methodology; Writing—original draft; Writing—review and editing. **Yilin Yu**: Investigation; Methodology. **Yinshi Ren**: Formal analysis; Investigation; Methodology. **Robert J Tower**: Formal analysis; Investigation; Methodology. **Guo-Fang Zhang**: Formal analysis; Investigation; Methodology. **Courtney M Karner**: Conceptualization; Formal analysis; Supervision; Funding acquisition; Investigation; Methodology; Writing—original draft; Writing—review and editing.

Source data underlying figure panels in this paper may have individual authorship assigned. Where available, figure panel/source data authorship is listed in the following database record: biostudies:S-SCDT-10_1038-S44319-024-00255-x.

## Disclosure and competing interests statement

The authors declare no competing interests.

# Expanded View Figures

**Figure EV1.  The transcriptomic and metabolic changes during osteoclastogenesis.**

(A, B) Gene Ontology pathway analysis of transcripts enriched in BMM (A) or mOC (B). (C) Volcano plots showing metabolites that differed significantly between BMM and mature osteoclast (mOC) ($n = 4$). Amino acids are colored red, and nucleotides are colored blue. (D) Metabolic pathway analysis (based on the Human Metabolome Database, HMDB) showing pathways enriched in mOC. (E–H) Graphical depiction of the intracellular abundance of select nucleotides in BMM, pOC, and mOC as determined by mass spectrometry ($n = 4$ independent experiments). (I) Metabolic pathways enriched in BMM. (J) Graphical depiction of the intracellular abundance of ATP ($n = 4$ independent experiments). (K–N) Graphical depiction of the intracellular abundance of select amino acids (alanine, glutamine, proline, and glutamate) in BMM, pOC, and mOC as determined by mass spectrometry ($n = 4$ independent experiments). Data are shown as mean ± SD. Two-tailed Student's unpaired $t$ test (A–D, and I). 1-way ANOVA (E–H, J, K–N).

▶

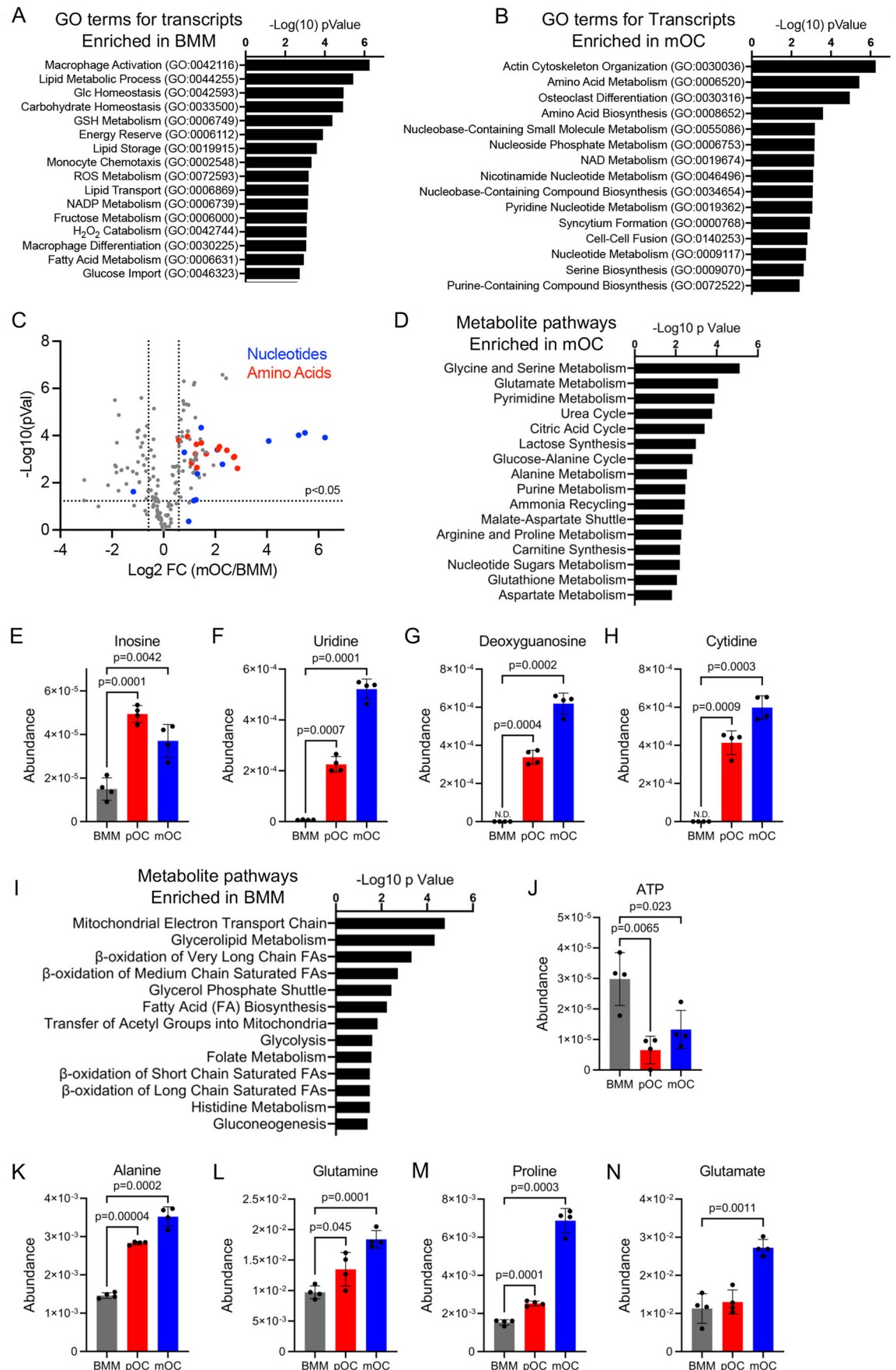

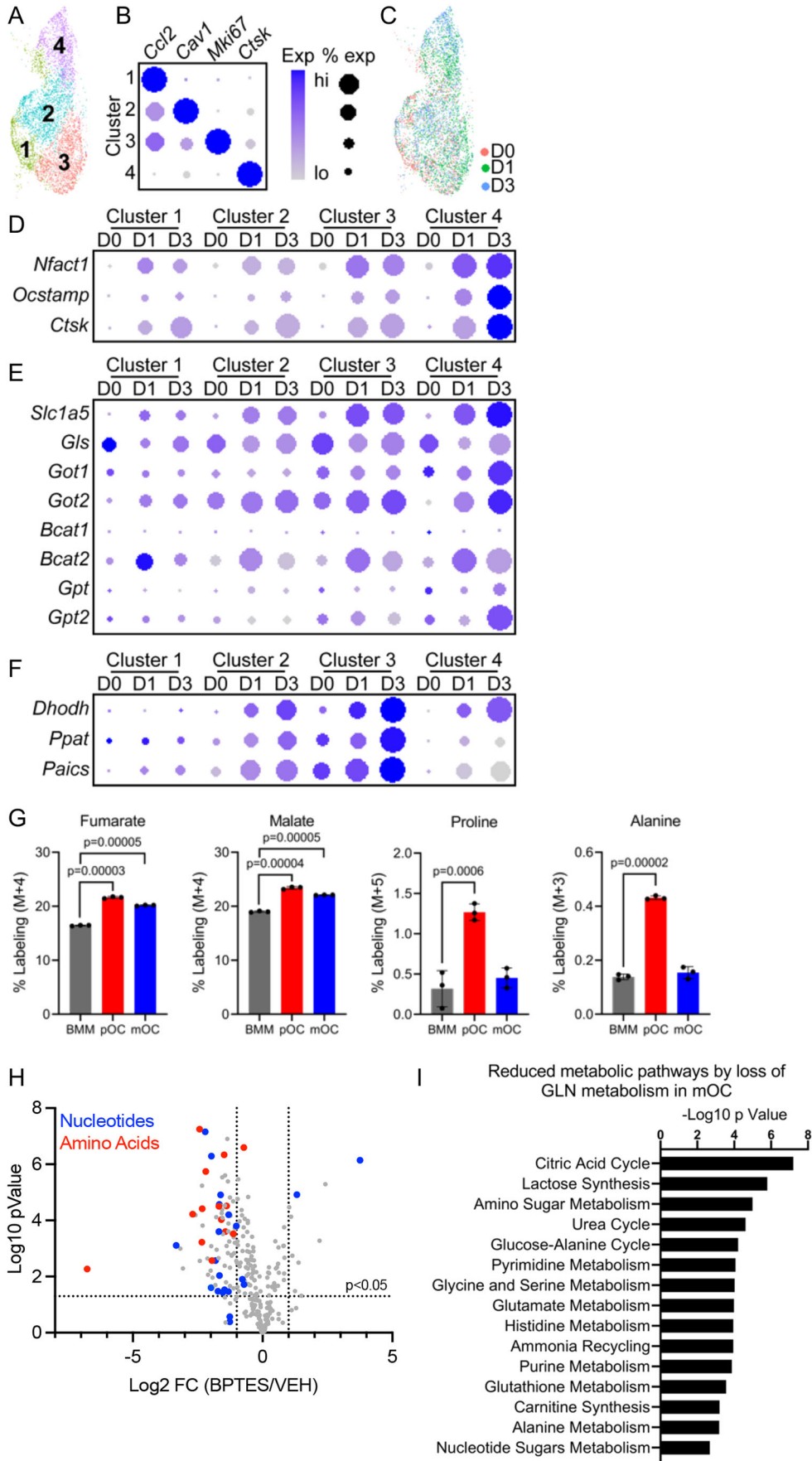

◀ **Figure EV2.  Glutaminolysis is activated during osteoclastogenesis to support metabolic reprogramming.**

(**A**) UMAP visualization of 4 clusters during osteoclastogenesis in the osteoclast culture system (Day 0–3 with RANKL). (**B**) Dot plot showing cluster marker genes. (**C**) UMAP visualization colored by days post RANKL stimulation. (**D**) Average osteoclast marker gene expression in 4 clusters. (**E**) The expression of genes associated with glutaminolysis in 4 clusters. (**F**) The expression of genes associated with nucleotide biosynthesis in 4 clusters. (**G**) Graphical depiction showing the percent labeling of fumarate, malate, proline, and alanine from [U-$^{13}$C]-glutamine in BMM, pOC, and mOC ($n = 3$ independent experiments). (**H**) Volcano plot showing metabolites that differed significantly between BPTES (10 μM)- and vehicle-treated mOC ($n = 4$). (**I**) Metabolic pathway analysis showing reduced pathways in BPTES-treated mOC. Data are shown as mean ± SD. One-way ANOVA (**G**). Two-tailed Student's unpaired $t$ test (**H**, **I**).

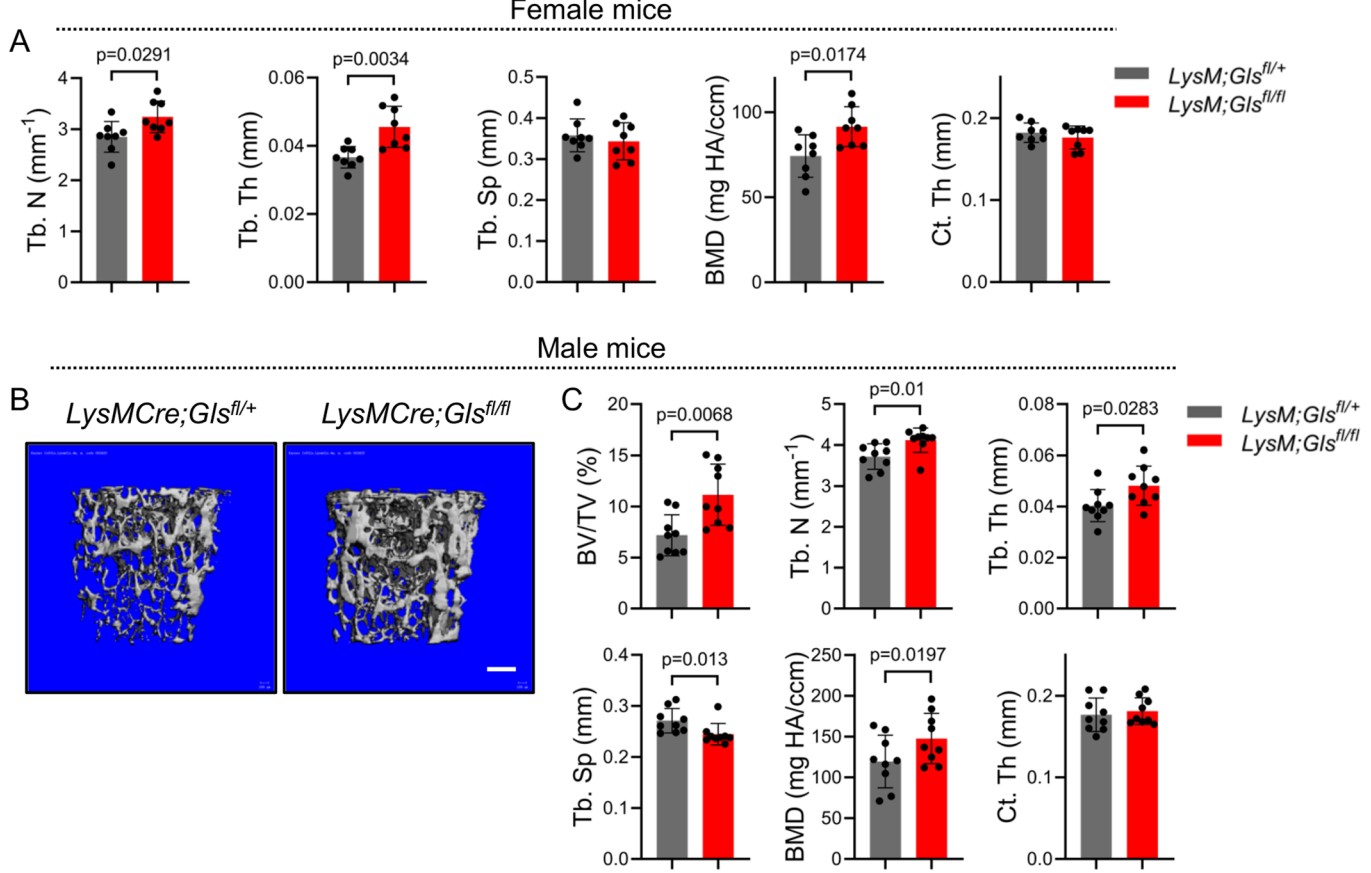

**Figure EV3. *Gls* ablation in myeloid lineage cells increases bone mass in both male and female mice.**

(A) µCT parameters of trabecular bone in the distal femurs of 4-month-old WT and *LysM;Gls^fl/fl^* female mice (*n* = 8 mice). (B, C) Representative µCT images (B) and µCT parameters of trabecular bone in the distal femurs of 4-month-old WT and *LysM;Gls^fl/fl^* male mice (*n* = 9 mcie). Scale bar: 200 µm. Data are shown as mean ± SD. Two-tailed Student's paired *t* test (A and C).

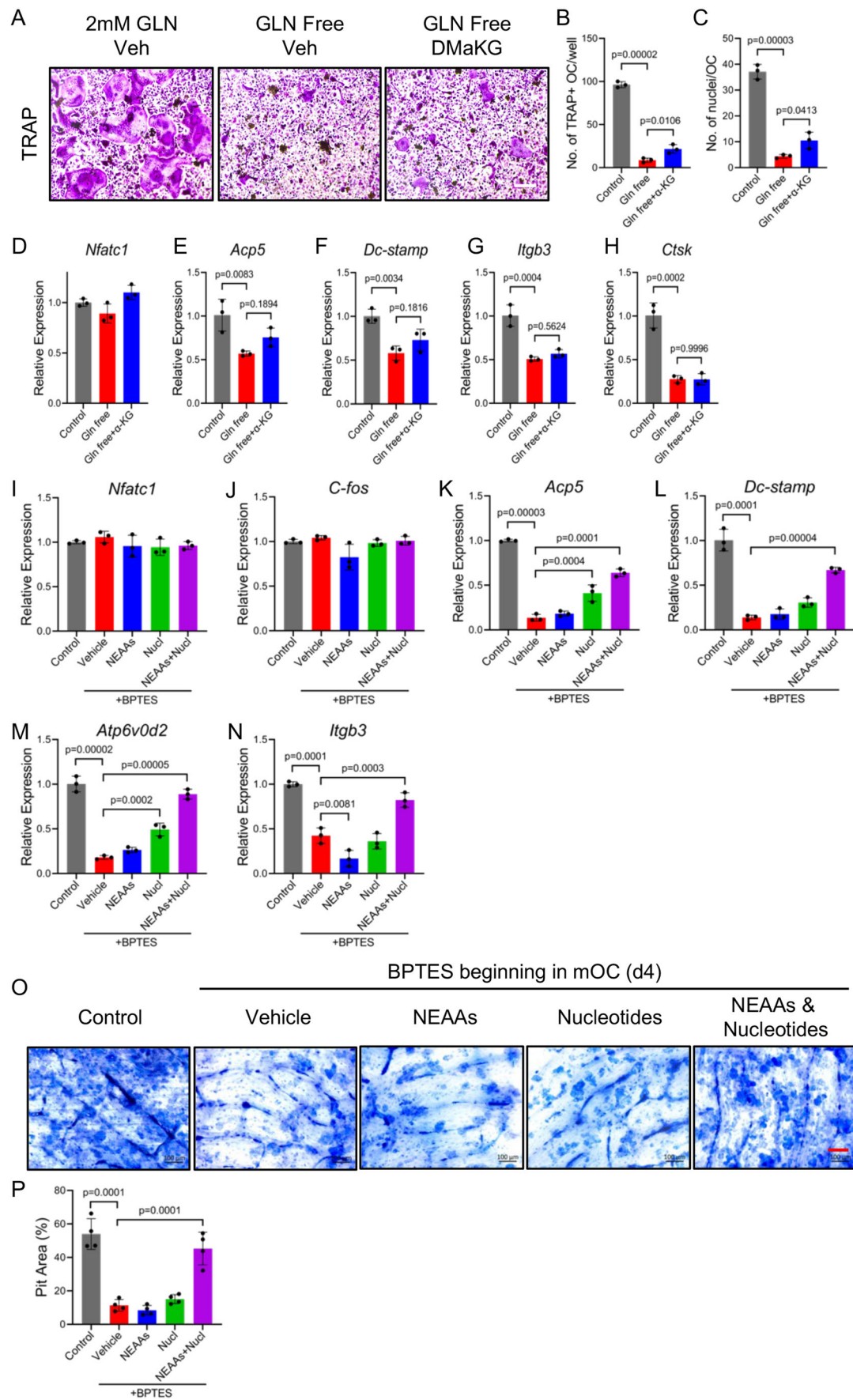

◀ **Figure EV4. Glutaminolysis provides amino acids and nucleotides, but not αKG, to regulate osteoclastogenesis.**

(A–H) The effect of supplementing dimethyl-α-ketoglutarate (DMαKG) on osteoclast differentiation as measured by TRAP staining (**A**, $n = 5$, scale bar: 500 μm) or qPCR analysis ($n = 3$ independent experiments) (**E–H**). (**C**) Quantification of TRAP-positive multi-nuclei cells from (**A**). (**D**) Quantification of the number of nuclei per TRAP-positive osteoclast from (**A**). (**I–P**) The effect of supplementing non-essential amino acids (NEAAs) and/or nucleotides (Nucl) on osteoclast differentiation and bone resorption as measured by qPCR analysis of osteoclast marker gene mRNA expression ($n = 3$ independent experiments) (**I–N**). (**O–P**) The effect of supplementing non-essential amino acids (NEAA) and/or nucleotides beginning at day 4 on bone resorption as measured by pit assay. Scale bar: 100 μm. (**P**) Quantification of the resorption pit area from **O** ($n = 4$ independent experiments). Data are shown as mean ± SD. One-way ANOVA (**B–H**), two-way ANOVA (**I–N, P**).

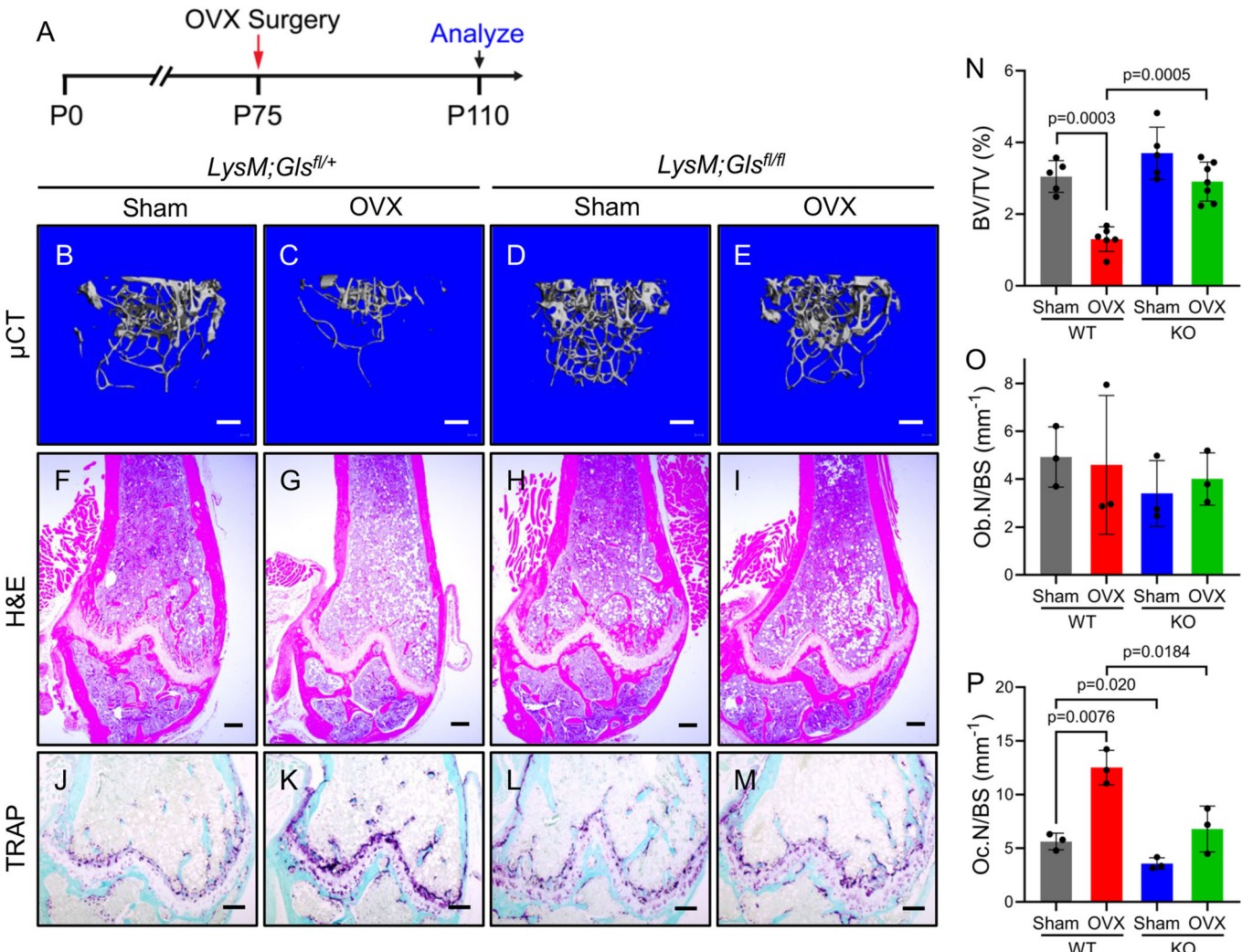

**Figure EV5.  Genetically targeting glutaminolysis prevents OVX-induced bone loss.**

(A) Schematic showing the experimental design. (B–M) Representative images of μCT (B–E, scale bar: 200 μm), H&E staining (F–I, scale bar: 100 μm), and TRAP staining (J–M, scale bar: 100 μm) of trabecular bone in the distal femurs of WT and *LysM;Gls^{fl/fl}* female mice subjected to OVX or sham surgery (*n* = 5–7). (N) The calculated BV/TV from μCT images (*n* = 5–7 mice). (O) Osteoblast number per bone surface (Ob.N/BS) quantified from H&E stains (*n* = 3 mice). (P) Osteoclast number per bone surface (Oc.N/BS) quantified from TRAP stains (*n* = 3 mice). Data are shown as mean ± SD. Two-way ANOVA (N–P).

