## [Peer Review File · EMBO Reports]

Glutaminolysis provides nucleotides and amino acids to regulate osteoclast differentiation in mice.

Guoli Hu, Yilin Yu, Yinshi Ren, Robert Tower, Guofang Zhang, and Courtney Karner

Corresponding author(s): Courtney Karner (Courtney.Karner@UTSouthwestern.edu)

Review Timeline:

Submission Date:	11th Jul 24
Editorial Decision:	25th Jul 24
Revision Received:	7th Aug 24
Accepted:	22nd Aug 24

Editor: Deniz Senyilmaz Tiebe

Transaction Report: The manuscript was transferred from another journal where it was originally reviewed. Since the original reviews are not subject to EMBO's transparent review process policy, they cannot be published.

Dear Dr. Karner,

Thank you for transferring your manuscript, which was previously revised and re-reviewed at another venue.

As discussed before, based on the referee comments, we would like to publish your manuscript pending minor revisions, where remaining minor concern of referee #1 needs to be addressed. Moreover, discussion points acknowledging the caveats pointed out by referee #3 regarding the osteoclast fusion/differentiation need to be included and the statements need to be toned down accordingly (points 1 and 4). All related publications pointed out by the referees need to be cited and discussed in the context of the current submission.

Moreover, I need you to address the points below before I can accept the manuscript (I apologize for the length of the list).

- Please provide 3-5 keywords for your study. These will be visible in the html version of the paper and on PubMed and will help increase the discoverability of your work.
- Please remove the figures from the manuscript text file.
- Please provide the referee reports and the point-by-point responses from the previous rounds of peer-review at the other venue.
- Currently, there is a section called 'Declaration of Interests' with multiple subsections. Of those subsections, - 'Data and Software availability' subsection needs to be its own section named as 'Data Availability'. Of note, the link provided in this section is invalid, which needs to be fixed. In this section, each dataset should be listed under a separate bullet point that includes 1) a short description of the measurement type (eg RNA-Seq, ChIP-Seq, mass spectrometry proteomics, imaging, etc...), 2) the name of the repository (or its recommended acronym, see table below and consult fairsharing.org); 3) the DOI or accession number of the dataset; and 4) a resolvable link to the dataset, either in the form of a resolvable link from <http://identifiers.org> or as the full URL to the respective database record. E.g.

The datasets and computer code produced in this study are available in the following databases:

RNA-Seq data: Gene Expression Omnibus GSE46843(<https://www.ncbi.nlm.nih.gov/geo/query/acc.cgi?acc=GSE46843>)

- The contents of 'Vertebrate animals' subsection needs to be included in the Methods section and the heading needs to be removed.

- 'Competing interest statement' subsection needs to be its own section and the title should be renamed as 'Disclosure Statement and Competing Interests'.

- Please remove the Author Contributions section from the manuscript.
- As per our format requirements, in the reference list, citations should be listed in alphabetical order and then chronologically, with the authors' surnames and initials inverted; where there are more than 10 authors on a paper, 10 will be listed, followed by 'et al.'. Please see <https://www.embopress.org/page/journal/14693178/authorguide#referencesformat>
- Please fill out and include an author checklist as listed in our online guidelines (<https://www.embopress.org/page/journal/14693178/authorguide>)
- The funding information should also be entered in our manuscript tracking system as it is acknowledged in the ms - an award from the Hamon Center for Regenerative Science and Medicine at UTSW.
- We noted the following regarding the figure callouts: Fig. 1J called out but we are unable to locate panel J in Fig. 1; Fig. 3A-F called out before Fig. 2E; Fig. 3I called out but we are unable to locate panel 3I; Fig. 6B-C called out before Fig. 5G-H; Fig. 7D-H called out before Fig. 6E; Fig. 7I-N and Fig. 7O-P called out, but the last panel in Fig. 7 is K; Supplemental Fig. 8D needs to be updated to the corrected callout; missing a callout for Fig. 6G and Suppl. Table 2. Please see <https://www.embopress.org/page/journal/14693178/authorguide#figureformat> for our figure nomenclature.
- We note that there are currently 10 Expanded View (EV) figures. Due to technical reasons, the maximum number of EV figures that we can accommodate is 5. Therefore, at least 5 of them and their legends should be part of the Appendix PDF file (nomenclature and callouts in the ms should be Appendix Figure S1, etc.); if the two suppl. Tables will be removed from the ms, they should be part of the same Appendix PDF (nomenclature and callouts in the ms should be Appendix Table S1, etc.)
- Supplementary Table 1 and 2: if these tables will remain in the manuscript, they need to be renamed as Table 1 and 2 (callouts included) and need to be placed between the main and EV figure legends. Alternatively, they can be a part of the Appendix (as mentioned in the above point).
- All Materials and Methods need to be described in the main text using our 'Structured Methods' format, which is required for all research articles. According to this format, the Methods section includes a Reagents and Tools Table (listing key reagents, experimental models, software and relevant equipment and including their sources and relevant identifiers) followed by a Methods and Protocols section describing the methods using a step-by-step protocol format. The aim is to facilitate adoption of the methodologies across labs. More information on how to adhere to this format as well as a downloadable template (.docx) for the Reagents and Tools Table can be found in our author guidelines: <https://www.embopress.org/page/journal/14693178/authorguide#structuredmethods>.

- The manuscript sections should be in the following order: Title page - Abstract & Keywords - Introduction - Results - Discussion - Methods - Data Availability - Acknowledgments - Disclosure Statement & Competing Interests - References - Figure Legends - (Main Tables with legends) - Expanded View Figure Legends.
- 'Experimental Model And Subject Details' section should be renamed as Methods.
- Please resubmit source data as one zip file per figure, so that there is a single .zip file per figure containing the separate source data files for the panels. This is necessary so that the data belonging to a specific figure is directly linked in the online version of the article.
- Our production/data editors have asked you to clarify several points in the figure legends:
 - o Please note that the extended data figures 7b-h, p is mislabeled as figure 7h, o in the statistical test information in the manuscript. This needs to be rectified.
 - o Please define the annotated p values *** as well as provide the exact p-values for the same in the legend of figure 7k; as appropriate.
 - o Please note that the exact p values are not provided in the legends of figures 1f, h-i; 2b, d-i; 3b, d-e; 5d; 6d-g; 7b-f; 8f, o; extended data figures 1e; 2a; 3g; 7b-c, k-n, p; 9c.
 - o Please indicate the statistical test used for data analysis in the legends of figures extended data figures 1a-b, d, i; 3i; 8c.
 - o Please note that information related to n is missing in the legends of figures 4c, f; 6d-g, extended data figures 8d; 9c.
 - o Although 'n' is provided, please describe the nature of entity for 'n' in the legends of figures 1e-i; 2b, d-i; 3b, d-f, h; 5c-d, f; 7b-i; 8f, o-p, u; extended data figures 1e-h, j; 2a-d; 3g; 5d-e; 6b-c; 7b-n, p; 10n-p.
 - o Please note that the scale bar is missing for figures 8b-d, g-i, extended data figures 10b-d, f-h, j-l.
- Papers published in EMBO Reports include a 'synopsis' and 'bullet points' to further enhance discoverability. Both are displayed on the html version of the paper and are freely accessible to all readers. The synopsis includes a short standfirst summarizing the study in 1 or 2 sentences (max 35 words) that summarize the paper and are provided by the authors and streamlined by the handling editor. I would therefore ask you to include your synopsis blurb and 3-5 bullet points listing the key experimental findings.
- In addition, please provide an image for the synopsis. This image should provide a rapid overview of the question addressed in the study but still needs to be kept fairly modest since the image size cannot exceed 550 (width) x 300-600 (height) pixels.

Thank you again for giving us to consider your manuscript for EMBO Reports, I look forward to your minor revision.

Kind regards,

Deniz Senyilmaz Tiebe

--

Deniz Senyilmaz Tiebe, PhD
Scientific Editor
EMBO Reports

Response to reviewers' comments (2nd review):**Reviewer #1:**

Thank you for your detailed revisions, which have strengthened the manuscript. One minor comment: related to Fig 4E and 6E – phalloidin staining shows peripheral actin belts on glass/plastic. Actin rings, important for formation of the sealing zone and resorptive lacuna, only occur in polarized OC on bone or hydroxyapatite substrates, and that is clearly not what was done here. Please revise the terminology.

We revised the manuscript to replace all instances of actin rings with actin belts.

Reviewer #2:

The authors addressed most of the concerns. A recent paper reported that GLS inhibitors have a dual impact on bone cells: they promote bone formation and inhibit the formation of cells that break down bone. Furthermore, two other studies (PMID: 30396165 and 38377021) showed that inhibiting GLS protected mice from OVX-induced bone loss. A new finding is that myeloid cell-specific GLS deletion affects OVX-induced bone loss, which has been moved to supplementary figures in the revised manuscript. Although the revised manuscript includes many new experiments, it mainly used GLS inhibitors. In addition, the study did not directly measure changes in the metabolites of cells lacking GLS, and the rescue experiments also used BPTES, a GLS inhibitor.

We have referenced these other studies as appropriate in the revised manuscript.

Reviewer #3 (Remarks to the Author):

In the revised manuscript, the authors performed additional experiments to demonstrate that glutaminolysis (not “glutamine” itself) is required for the osteoclast fusion.

The authors showed that treatment with glutamine derived metabolites (e.g., amino acids and nucleotides) could rescue the impaired osteoclast fusion in GLS-inhibited cells. This data is interesting and key to discriminate the importance of “glutamine metabolism” from glutamine itself.

Although the revisions the authors have made significantly improved the manuscript, there are still critical problems and inconsistencies.

1) The authors referred to the current two papers (Rohatgi et al., Nature Communications 2023, Stegen et al., Nature Metabolism 2024) and discussed the complementary nature among the authors' findings and previous reports, but the reviewer still feels that the conceptual advance of this paper over previous reports is limited to the hypothesis that glutamine metabolism is specifically required for the osteoclast fusion but not differentiation. However, the authors failed to provide sufficient evidence to support this hypothesis. The myeloid-specific GLS KO mice decreased osteoclast number, suggesting that osteoclast differentiation was affected in these animals (revised Fig.4). Mice with impaired osteoclast fusion (e.g. Dc-stamp KO mice, Oc-stamp KO mice) displayed an increased number of bone-resorbing mononuclear TRAP-positive cells,

and bone mass was not decreased in such animals (Yagi et al., JEM 2005, Miyamoto et al., JBMR 2012). The myeloid-specific GLS KO mice displayed a typical phenotype of mice with impaired osteoclast differentiation, but not osteoclast fusion.

We have softened the language in the revised manuscript. Specifically, we agree with the reviewer that the effects of inhibiting glutaminolysis specifically affects osteoclast differentiation. Thus, we have changed the text referring to defects in osteoclast fusion defects in osteoclast differentiation to be more precise.

2) The authors also failed to clarify the molecular mechanisms underlying how GLS regulates osteoclast fusion. The authors showed that inhibition of GLS decreased the expression of fusion genes (Dc-stamp and Atp6v0d2) without affecting Nfatc1 (revised Fig.3C, D). Since these genes are under the control of Nfatc1 (Kim et al., Mol Endocrinol 2008 PMID: 17885208), it is difficult to understand how GLS can regulate fusion genes independent of NFATc1.

3) Although the new data showing the rescue experiments (revised Fig.6) are interesting, the mode of action of glutamine derived metabolites (e.g., amino acids and nucleotides) in regulation of osteoclast fusion is still unclear.

In response to points 2 & 3, we currently don't understand the precise molecular mechanism by which GLS regulates osteoclast fusion and differentiation independent of Nfatc1. We have updated the discussion (p16, lines 319-323) as follows to reflect this uncertainty.

“Interestingly, inhibiting glutaminolysis specifically limited the induction of genes associated with osteoclast fusion (e.g., *Dc-stamp* and *Atp6v0d2*) and terminal differentiation (e.g., *Catk*) without affecting NFATC1 expression. This is peculiar as these genes are directly regulated by NFATC1 (Kim et al, 2008; Matsumoto et al, 2004). The effects of inhibiting glutaminolysis on osteoclast differentiation, fusion and function are likely multifaceted. We posit that reducing amino acid and nucleotide synthesis are the prime culprits...”

4) In the abstract, the authors stated “glutaminolysis is both necessary and sufficient for osteoclastogenesis and bone resorption in vivo”. This is clearly an overstatement and there is no evidence showing that glutaminolysis is sufficient for osteoclastogenesis.

We agree with the reviewer and have updated that sentence in the abstract as follows:

“... glutaminolysis is essential for osteoclastogenesis and bone resorption *in vivo*.”

Dr. Courtney Karner
University of Texas Southwestern Medical Center
5323 Harry Hines Blvd
F5.102A
Dallas, TX 75235
United States

Dear Dr. Karner,

Thank you for submitting your revised manuscript. I have now looked at everything and all is fine. Therefore, I am very pleased to accept your manuscript for publication in EMBO Reports.

Congratulations on a nice work!

Kind regards,

Deniz Senyilmaz Tiebe

--

Deniz Senyilmaz Tiebe, PhD
Senior Scientific Editor
EMBO Reports

--
